# Intermolecular channels direct crystal orientation in mineralized collagen

YiFei Xu [1,2,3], Fabio Nudelman [1,4], E. Deniz Eren[2,5], Maarten J. M. Wirix[1,2], Bram Cantaert[3],
Wouter H. Nijhuis[6], Daniel Hermida-Merino[7], Giuseppe Portale [7,8], Paul H. H. Bomans[1,2],
Christian Ottmann[2,9], Heiner Friedrich [2,5], Wim Bras [7,10], Anat Akiva [1,2,11], Joseph P. R. O. Orgel[12✉],
Fiona C. Meldrum [3✉] & Nico Sommerdijk [1,2,13✉]

The mineralized collagen fibril is the basic building block of bone, and is commonly pictured as a parallel array of ultrathin carbonated hydroxyapatite (HAp) platelets distributed throughout the collagen. This orientation is often attributed to an epitaxial relationship between the HAp and collagen molecules inside 2D voids within the fibril. Although recent studies have questioned this model, the structural relationship between the collagen matrix and HAp, and the mechanisms by which collagen directs mineralization remain unclear. Here, we use XRD to reveal that the voids in the collagen are in fact cylindrical pores with diameters of ~2 nm, while electron microscopy shows that the HAp crystals in bone are only uniaxially oriented with respect to the collagen. From in vitro mineralization studies with HAp, $CaCO_3$ and $\gamma$-FeOOH we conclude that confinement within these pores, together with the aniso-tropic growth of HAp, dictates the orientation of HAp crystals within the collagen fibril.

---

[1] Department of Chemical Engineering and Chemistry, Laboratory of Materials and Interface Chemistry and Center for Multiscale Electron Microscopy, Eindhoven University of Technology, PO Box 513, 5600 MB Eindhoven, The Netherlands. [2] Institute for Complex Molecular Systems, Eindhoven University of Technology, PO Box 513, 5600 MB Eindhoven, The Netherlands. [3] School of Chemistry, University of Leeds, Woodhouse Lane, Leeds LS2 9JT, UK. [4] School of Chemistry, University of Edinburgh, Joseph Black Building, The King's Buildings, David Brewster Road, Edinburgh EH9 3FJ, UK. [5] Department of Chemical Engineering and Chemistry, Laboratory of Physical Chemistry, Eindhoven University of Technology, PO Box 513, 5600 MB Eindhoven, The Netherlands. [6] Department of Orthopaedic Surgery, University Medical Centre Utrecht, Wilhelmina Children's Hospital, Utrecht, The Netherlands. [7] Netherlands Organization for Scientific Research (NWO), DUBBLE@ESRF, BP220, F38043 Grenoble, France. [8] Macromolecular Science and New Polymeric Materials, Zernike Institute for Advanced Materials, University of Groningen, Nijemborg 4, 9747 Groningen, The Netherlands. [9] Department of Biomedical Engineering, Eindhoven University of Technology, PO Box 513, 5600 MB Eindhoven, The Netherlands. [10] Chemical Sciences Division, Oak Ridge National Laboratory, One Bethel Valley Road, Oak Ridge, TN 37831, USA. [11] Department of Cell Biology, Radboud Institute of Molecular Life Sciences, Radboud University Medical Center, Geert Grooteplein, 6525 GA Nijmegen, The Netherlands. [12] Departments of Biology, Physics and Biomedical Engineering, Pritzker Institute of Biomedical Science and Engineering, Illinois Institute of Technology, Chicago, IL 60616, USA. [13] Department of Biochemistry, Radboud Institute of Molecular Life Sciences, Radboud University Medical Center, Geert Grooteplein, 6525 GA Nijmegen, The Netherlands. ✉email: orgel@iit.edu; f.meldrum@leeds.ac.uk; nico.sommerdijk@radboudumc.nl

Bone possesses exceptional mechanical properties, where these derive from its complex composition and structure. Principally comprising plate-like carbonated hydroxyapatite (HAp) crystals and collagen fibrils, bone is organized over up to nine hierarchical levels[1–3], where the HAp crystals efficiently bear the stress applied to the collagen, increasing its stiffness, fracture strength and robustness[4–7]. The HAp crystals are located both within (intrafibrillar) and around (extrafibrillar) the fibrils[2,3,8], where the intrafibrillar crystals have received the lion's share of attention. The organization of these crystals is usually depicted according to the traditional model of a 3-dimensional deck-of-cards structure in which the platelets lie morphologically and crystallographically parallel throughout a fibril (Fig. 1)[1,9–14]. However, recent studies have questioned the validity of this model, providing evidence that the mineral platelets only form small stacks of 2–4 platelets[2], and that they exhibit uniaxial orientation with respect to the collagen long axis[15,16].

The mechanism by which collagen directs the orientation of these intrafibrillar HAp crystals is also a topic of debate[15,17–19]. HAp forms within specific gap regions inside the collagen fibrils (also called the hole zone) which contain rectangular (2D) channels[1,12,13,20] that have long been thought to both induce nucleation[9,21–23] and provide an organized organic matrix that guides epitaxial growth (Fig. 1)[9,13,18,21]. This mechanism has been recently called into question, however, by the demonstration that the HAp platelets in bone are covered by a hydrated amorphous layer that would preclude such molecular recognition[24]. As an alternative, Gower and coworkers suggested that the orientation of the crystals may be governed by their confinement within the collagen fibrils[15], which was supported by the demonstration that cylindrical nanopores can direct the oriented growth of the HAp crystals in vitro[25]. However, this suggestion could not be considered further without a detailed model of the ultrastructure of the gap region in collagen. Existing models also do not explain the longstanding observation that the HAp nanoplates start as needle-like crystals[2,12], and only later develop their 2D shape.

To resolve the mechanism by which collagen directs the oriented growth of HAp crystals, we first revisit the orientation of these crystals within the collagen fibrils in bone, as well as the shape of the gap zone in the unmineralized collagen. Detailed electron microscopic analysis shows unequivocally that mineralized single collagen fibrils in human bone indeed contain small stacks of a few crystals, and that the mineral platelets only exhibit uniaxial orientation. Using X-ray diffraction (XRD) we then visualize the 3D structure of the gap regions of the unmineralized collagen fibril and demonstrate the presence of elongated cylindrical nanopores which we propose direct HAp growth, as previously demonstrated for in vitro model systems. Finally, in vitro mineralization of the fibrils with alternative, non-native minerals shows that this orientation mechanism is only possible thanks to the strongly anisotropic structure of HAp. This provides compelling evidence that the orientation of HAp within collagen fibrils arises due to confinement effects only.

## Results

**Electron tomographic analysis of mineralized collagen in bone.** To resolve the actual orientation of the apatite crystals in human bone, we performed a high-resolution electron tomography (ET) study (Fig. 2, see also Supplementary Movie 1) on waste material from a surgical procedure on a fractured tibia from a 10-year-old healthy female. Energy-filtered TEM tomography was used to visualize both the collagen fibrils and the HAp platelets. Using focused ion beam (FIB) milling, a 100 nm-thick lamella was cut parallel to the long axes of the mineralized collagen fibrils, and analyzed by low dose ET (total dose = 65 e$^-$ Å$^{-2}$, see also Supplementary Fig. 1. Computer generated lateral tomography reconstruction slices viewed perpendicular to the long axes of the collagen fibrils are defined as $y$-slices, while the longitudinal reconstructed slices viewed along the fibril long axes are defined as $z$-slices)[26].

To optimize contrast for the visualization of the mineralized collagen fibrils, 70 $y$-slices (slice thickness 0.75 nm) of the tomographic reconstruction were averaged to generate a 2D projected volume with a thickness of 52.5 nm (Fig. 2). The resulting 2D projection showed collagen fibrils with diameters of ~100 nm (Fig. 2a) that displayed the characteristic ~67 nm banding pattern. The collagen fibrils are closely packed and heavily mineralized with HAp crystals, where the discontinuity in the banding pattern in the lateral direction allowed the individual fibrils to be recognized. The precise boundaries of the fibrils were identified by viewing the reconstructed tomogram along the fibril axis (Fig. 2e). To optimize the contrast for visualization of the organic fibrils, 400 $z$-slices (slice thickness 0.75 nm) of the tomogram were added, generating a 2D projected volume with a thickness of 300 nm.

Low dose selected area electron diffraction (LDSAED, Fig. 2d) of an area containing two parallel fibrils (Fig. 2c) shows the characteristic pair of (002) diffraction arcs, which indicates that the HAp platelets are oriented with their $c$-axes along the fibrils. In line with previous reports of bone and in vitro mineralized collagen[15,27], an additional diffraction ring was observed at ~2.8 Å, which can be attributed to the unresolved (112), (211), and (300) diffractions. These reflections arise from multiple lattice planes, and can only co-exist when the planes of the HAp platelets are misoriented by >60°. This implies that the platelets are only uniaxially oriented[15]. No other calcium phosphate phase was detected, although it is possible that the sample still contains

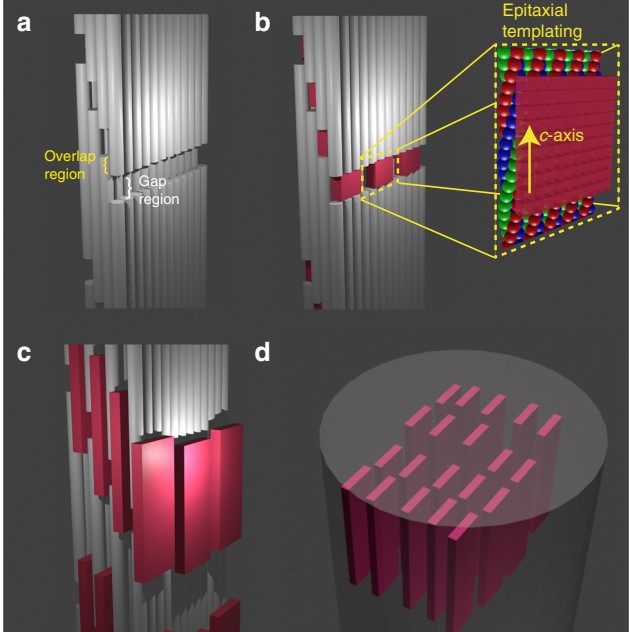

**Fig. 1 Schematic representation of the deck-of-cards model of mineralized collagen.** The model considers the collagen fibril to comprise staggered molecular layers and the gap regions to contain rectangular shaped channels (**a**)[1,11,13]. In that model the HAp crystals nucleate in the gap regions (**b**), where they are epitaxially templated by charged functional groups (inset in **b**)[9,18]. They subsequently develop into platelets that are embedded between the collagen layers as shown in **c**. A 3D array of oriented parallel platelets eventually forms, as shown in **d**.

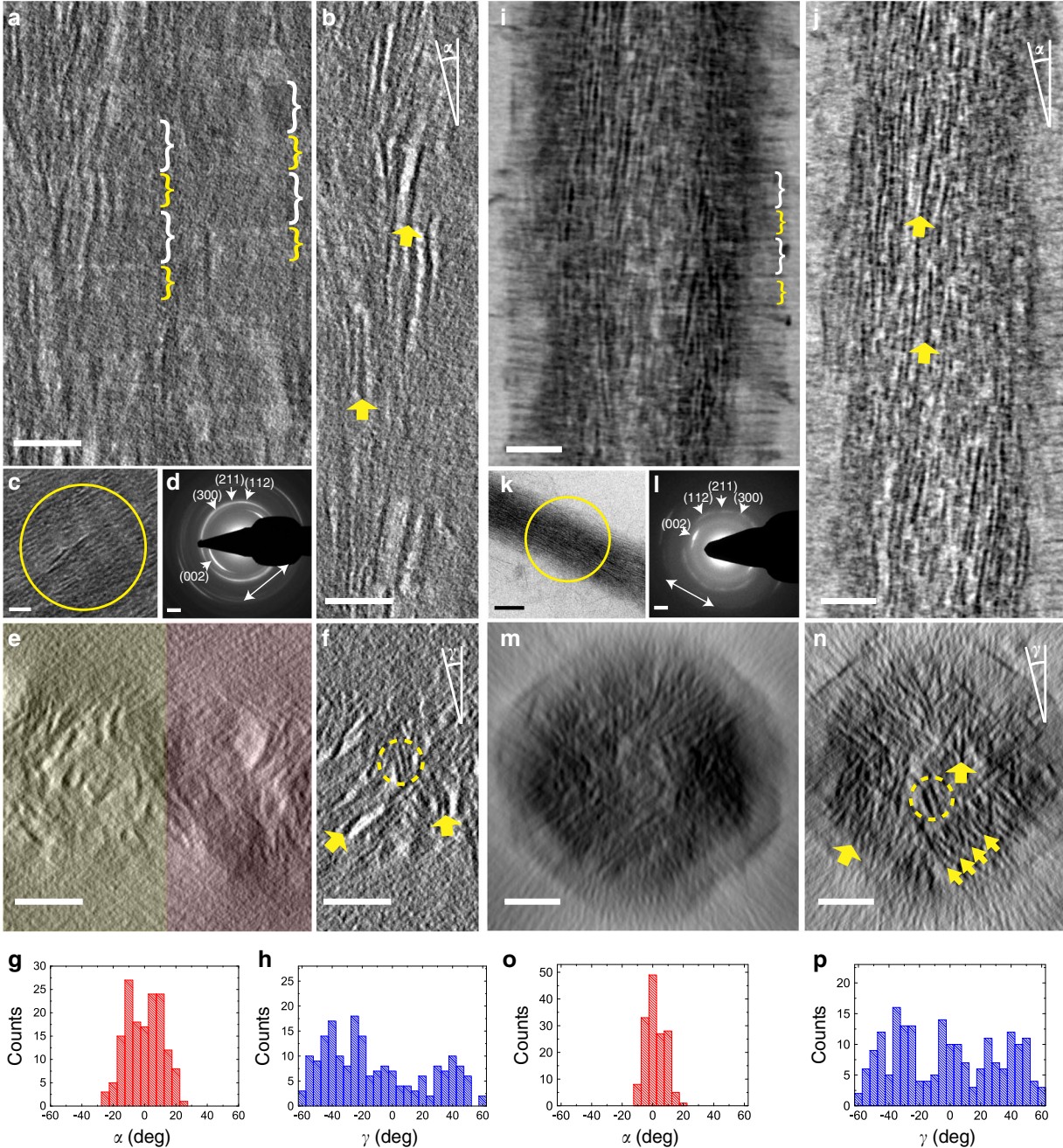

**Fig. 2 Electron tomography of in vitro and in vivo mineralized collagen fibrils. a–h** Electron tomography of a human bone lamella. **a** 2D projection of 70 tomographic reconstruction *y*-slices showing two adjacent mineralized collagen fibrils. **b** Tomographic reconstruction *y*-slice of one single fibril averaged from five adjacent slices. **c** TEM image of two parallel fibrils. **d** LDSAED pattern of the circled area in **c**. **e** 2D projection of 400 cross-section tomographic reconstruction *z*-slices showing two adjacent fibrils, as highlighted in yellow and red, respectively. **f** Tomographic reconstruction *z*-slice of one single fibril averaged from 10 adjacent slices. **g, h** Longitudinal and lateral angular distributions ($\alpha$ and $\gamma$, as indicated in **b** and **f**, respectively) of HAp platelets in the fibril. **i–p** Cryogenic electron tomography of a type I collagen fibril mineralized with HAp in the presence of pAsp and stained with uranyl acetate[38]. **i** 2D projection of 50 tomographic reconstruction *y*-slices showing one mineralized collagen fibril. **j** Tomographic reconstruction *y*-slice of one single fibril averaged from five adjacent slices. **k** CryoTEM image of one single fibril. **l** LDSAED pattern of the circled area in **k**. **m** 2D projection of 300 tomographic reconstruction *z*-slices of one single fibril. **n** Tomographic reconstruction *z*-slice of one single fibril averaged from 10 adjacent slices. **o, p** Longitudinal and lateral angular distributions of the platelets, respectively. The gap/overlap regions are highlighted in **a** and **i** by white/yellow brackets, respectively. Gap regions in **a** are darker than overlap regions due to heavy metal staining (see Supplementary Fig. 1). Long axis of the fibrils are indicated by white arrows in **d** and **l**. The orientations of some HAp platelets are highlighted in **b**, **f**, **j**, and **n** by yellow arrows. Stacks of HAp platelets are highlighted in **f** and **n** by yellow circles, while a stack of ~10 HAp platelets is highlighted in **n** by an array of yellow arrows. ~150 crystals were measured that were at least 20 nm apart for **b** and **j**, and 100 nm apart for **f** and **n**. Scale bars: all panels 50 nm, except **d** and **i** 2 nm$^{-1}$.

small amounts of amorphous calcium phosphate (ACP) and octacalcium phosphate (OCP). These have been suggested to be precursors to bone mineralization[28–30] and can still exist in mature bone[24,31], but are difficult to identify by LDSAED.

Most crystals were intrafibrillar with dimensions of ~65 × 20 × 3 nm (Fig. 2b and f, see also Supplementary Fig. 2), and sometimes displayed a curved, propeller-like morphology as previously reported (Supplementary Fig. 3)[2]. A smaller number of extrafibrillar crystals were also observed near the boundaries of the fibrils, where these had the same size and shape as the intrafibrillar crystals, and usually propagated into the fibrils (Supplementary Fig. 3). Analysis of the orientations of the intrafibrillar crystals confirmed that they were aligned along the long axis of the collagen fibril, with an angular distribution of ~ ±20° (Fig. 2g, see also Supplementary Fig. 2 and Supplementary Movie 2). The ± values here and in later discussions all correspond to the maximum misalignment angles, based on the analysis of ~150 crystals.

Definitive proof for the uniaxial orientation of these crystals then came from viewing the tomographic reconstruction along the collagen fibril axis. 2D projections of 10 added adjacent tomographic z-slices (total slice thickness 7.5 nm) showed that the platelets are indeed not organized in an ordered deck of cards structure (Fig. 2f, see also Supplementary Movie 3). Instead, in most regions of the fibril the z-slices showed that the crystals had random orientations (Fig. 2h) forming small stacks of just 2–4 platelets. In areas with higher mineral density, larger stacks of ~8 platelets were occasionally observed, (see Supplementary Fig. 4) but these were still randomly oriented within the diameter of the collagen fibril (~100 nm). Significantly, as the investigated volumes only had a thickness of 7.5 nm along the fibril axis, the observed different crystal orientations are independent of the twisted structure of the embedding collagen fibrils[27]. Needles (~2 nm in width and ~10 nm in length, Supplementary Figs. 4 and 5) were observed at the tip of some platelets, which confirms previous observations[2,12] and supports the hypothesis that the HAp platelets evolve from needle-like crystals.

The observed uniaxial orientation of the HAp crystals seems to contrast with the pioneering tomography work of Landis et al. in the 1990s on mineralized turkey tendon[13] and chick bone[1], where that data is consistent with the deck-of-cards model. We note, however, that this difference may derive from their use of thicker samples (250 and 500 nm, respectively) and of a less sensitive camera system. Both of these limit the detection of HAp platelets that are not oriented parallel to the electron beam, as platelets oriented edge-on will have significantly higher contrast in TEM imaging. This was confirmed by analysis of a model system starting from different image qualities (see Supplementary Figs. 6 and 7 for details).

**X-ray analysis of the structure of unmineralized collagen fibrils**. To understand the role of collagen in directing the oriented growth of the HAp crystals, we examined the structure of the collagen gap zone by analyzing the 3D electron density map created from existing X-ray data of hydrated type-I collagen fibrils[32]. The data show that the collagen molecules are organized in a triclinic superstructure, with only a modest degree of lateral organization, and without real long-range order. This super-structure consists of unit cells called microfibrils, which each comprise five 1D-staggered, twisted collagen triple helix molecules. In the overlap region of this structure, the collagen molecules are tightly packed in a quasi-hexagonal manner with a tilt relative to the c-axis of the unit cell and an intermolecular spacing of <1 nm (Fig. 3a and b). The widths of the collagen triple helix molecules vary between ~1.1 and ~1.5 nm, in line with previous

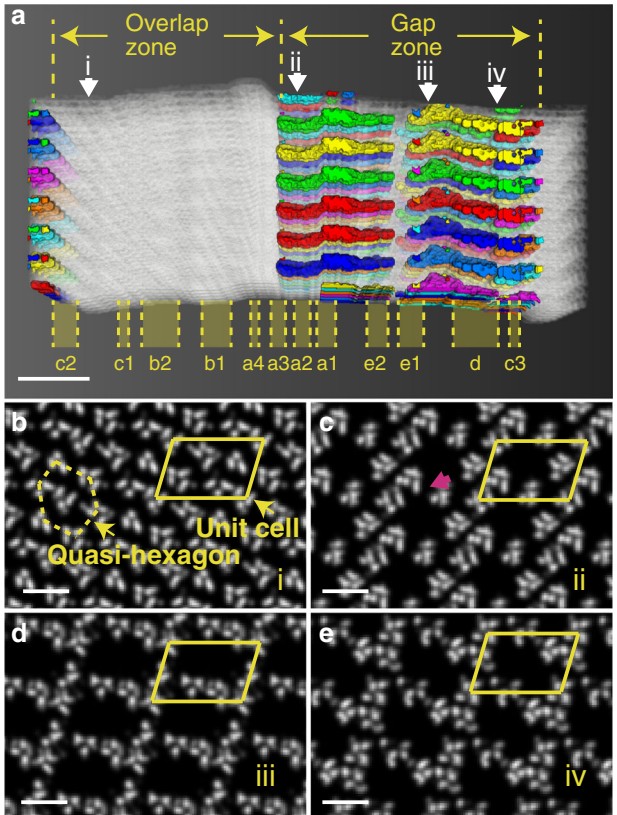

**Fig. 3 Analysis of the 3D electron density map of the structure of collagen.** The electron density map is created from X-ray diffraction data[32]. **a** Intermolecular voids within the collagen structure, with the gap/overlap zones and the banding structure labeled. The white structures correspond to the collagen molecules, and the channels between them are labeled with different colors based on their connectivity. **b–e** Cross-section slices viewed along the collagen fibril cut at position i (0.075 D), ii (0.505 D), iii (0.773 D) and iv (0.872 D) as highlighted by white arrows in **a**, respectively. **b** Shows the typical structures of overlap region, while **c–e** show the 2–3 nm wide channels in gap region with varying cross section shapes. The unit cell is highlighted by the yellow parallelogram. A ~0.3 nm wide small connection is highlighted by magenta arrow in **c**. Scale bars: **a** 2 nm, **b–e** 10 nm.

reports[30,33,34]. In the gap region, however, the collagen molecules are oriented approximately parallel to the unit cell c-axis (slightly tilted in the opposite direction to that of the overlap tilt). Importantly, one of the five collagen molecules is missing in each unit cell due to the 1D-staggering and the non-integer dividend of the collagen molecules length and D. As a consequence, each collagen unit cell contains a straight channel which runs parallel to the long axis of the fibril (Fig. 3a, and Supplementary Movie 4). Each of the channels is 2–3 nm in width and its cross section varies in shape along the channel (as shown by Fig. 3c–e, and Supplementary Movie 4). The channels span the gap zone and have a discontinuity between the so-called e2 and e1 bands (Fig. 3a)[35], which separates them into discrete sections of 15–20 nm in length. In lateral directions, these channels are mostly separated from each other, with only ~0.3 nm wide small connections (Fig. 3c, magenta arrow). These channel geometries are consistent with recent computational data[36], and are mark-edly different from the traditional picture which depicts rectangular 2D channels in the gap region (Fig. 1)[1,13,19,37].

**Cryo-TEM analysis of collagen fibrils in vitro mineralized with HAp**. To understand how these cylindrical channels can direct

the crystal orientation of HAp in bone, we revisited our minimal in vitro model system. There, self-assembled type I collagen fibrils from horse tendon were mineralized by immersing them in a solution containing $CaCl_2$, $K_2HPO_4$ and poly(aspartic acid) (pAsp) for 72 h (see "Methods" section for details)[38], where the pAsp was used as a crystallization control agent following the method developed by Gower et al. [15,16]. The fidelity of this model system was certified using cryo-ET[39] to carry out a detailed comparison of the size, shape, and orientation of the intrafibrillar crystals with those observed in bone.

Analysis of existing cryo-electron tomograms[38] revealed intrafibrillar platelets with dimensions of $60 \times 15 \times 4$ nm (Figs. 2i–p, see also Supplementary Fig. 2), where these are comparable to the dimensions previously reported for HAp crystals in human bone (thickness 2–8 nm, width 20–30 nm, length 50–100 nm)[2,40–42], and to those derived from our own data (Fig. 2a–h). The orientation of the crystallites was also similar to those in the human bone sample. The 3D reconstruction of the cryo-electron tomogram showed that the intrafibrillar platelets were aligned with the fibril axis (Fig. 2j and o, see also Supplementary Movie 5), while the HAp (002) reflections appeared as a pair of narrow arcs with an angular spread of ±15° (Fig. 2k and l). The reflections corresponding to the (112), (211), and (300) planes were also observed as three arcs with comparable $d$-values (~2.8 Å) at angles of 36°, 66°, and 90° with respect to the (002) diffraction signal[15,27], again pointing to uniaxial orientation of the crystals. The tomogram further showed that adjacent HAp platelets form small stacks of up to 10 platelets (Fig. 2n and Supplementary Movie 6), and that these had no preferred lateral orientation (Fig. 2p). The structure is therefore comparable to that observed in the human bone sample (Fig. 2f and h)[2,19]. Notably, we also obtained similar crystal sizes and orientations for the in vitro mineralization of type I collagen fibrils separated from bovine Achilles tendon (Supplementary Fig. 8)[15], demonstrating that these results are not specific to the source of the collagen.

The above detailed comparison between the in vivo and in vitro results shows that our minimal model system provides a reliable means of studying the intrafibrillar mineralization in bone. Further, as pAsp was the only non-collagenous component present, it additionally indicates that the uniaxial orientation of intrafibrillar HAp crystals does not require extensive biological control, but can be attributed to the initial confinement within the intermolecular channels inside the collagen fibril. It is also important to point out that HAp crystals precipitated in bulk solution have similar thicknesses to those formed within the collagen fibril, but are about twice as wide and long (Supplementary Fig. 9). These crystals express the same crystallographic faces as the biogenic crystals, indicating that their morphology is related to the surface energies of the expressed crystal planes under the solution conditions used, rather than to specific interactions with biological macromolecules.

**Mineralization of collagen with non-native minerals**. To confirm whether the orientation of intrafibrillar minerals derives from specific interactions between the mineral and the amino acid residues within the gap region—as previously proposed[9,13,18,21]—or whether it is just due to confinement of the mineral in the intermolecular channels, we mineralized collagen with lepidocrocite ($\gamma$-FeOOH) and calcium carbonate. These two mineral systems offer different structures and compositions and have no relevance to the mineralization of collagen in biological systems.

Type I collagen fibrils (bovine Achilles) were mineralized with $\gamma$-FeOOH by immersing them in a solution of $FeCl_2$, $FeCl_3$, and pAsp, and increasing the pH by exposure to ammonia (Supplementary Fig. 10)[43]. Analysis of fibrils isolated after 3 weeks revealed the presence of $\gamma$-FeOOH platelets with average

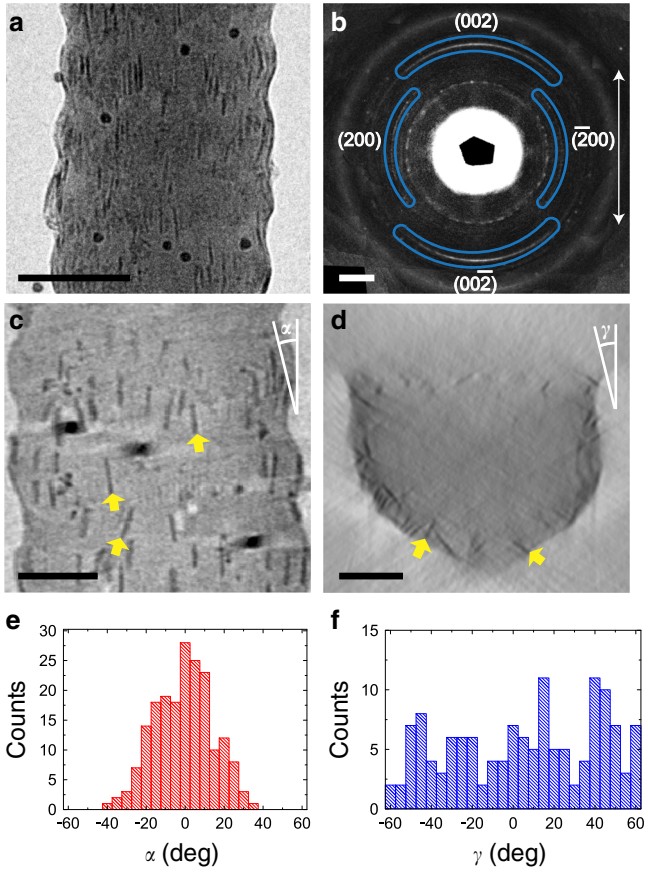

**Fig. 4 Type I collagen fibril mineralized with $\gamma$-FeOOH in the presence of pAsp. a** TEM image of a collagen fibril containing $\gamma$-FeOOH platelets. **b** LDSAED of **a** showing wide arcs of $\gamma$-FeOOH (002) diffraction along the collagen fibril axis. The pattern is obtained after radial alignment and averaging of 9 individual patterns (see Supplementary Figs. 13 and 14, and Supplementary Table 1). The direction of the collagen fibril long axis is highlighted by the white arrow. **c**, **d** Tomographic reconstruction $y$- and $z$-slices, respectively. **d** Is averaged from 10 adjacent slices to enhance the contrast. The orientations of some $\gamma$-FeOOH platelets are highlighted by yellow arrows in **c**, **d**. **e**, **f** Longitudinal and lateral angular distribution ($\alpha$ and $\gamma$, as indicated in **c** and **d**) of the platelets, respectively. ~150 crystals were measured that were at least 10 nm apart for **e** and 50 nm apart for **f**. Scale bars: **a** 100 nm, **b** 2 nm$^{-1}$, **c**, **d** 50 nm.

dimensions of ~$25 \times 13 \times 2.5$ nm adjacent to the fibril surface only (Fig. 4a and d), where shallow infiltration is attributed to the low solubility of $Fe^{2+}/Fe^{3+}$ ions in alkaline solutions[44]. In contrast, $\gamma$-FeOOH crystals with average dimensions of ~$77 \times 25 \times 3$ nm formed outside the fibrils under the same conditions. These were longer and wider than the intrafibrillar crystals, but in common with HAp had comparable thicknesses and were elongated along the $c$-axis (Supplementary Fig. 11)[45]. No intrafibrillar mineralization occurred in the absence of pAsp, but instead the fibrils became coated with 5–10 nm ferrihydrite particles (Supplementary Fig. 12).

LDSAED patterns of a single mineralized fibril exhibited a pair of broad arcs with angular spreads of ~80°, corresponding to the (002) planes of $\gamma$-FeOOH (Fig. 4b, see details in Supplementary Figs. 13 and 14 and Supplementary Table 1). The centers of the arcs, and therefore the $c$-axes of these platelets, were aligned with the long axis of the fibrils. ET also showed that the long axes of the platelets were generally oriented with the fibril axis (Fig. 4c and Supplementary Movie 7), but with a wider angular

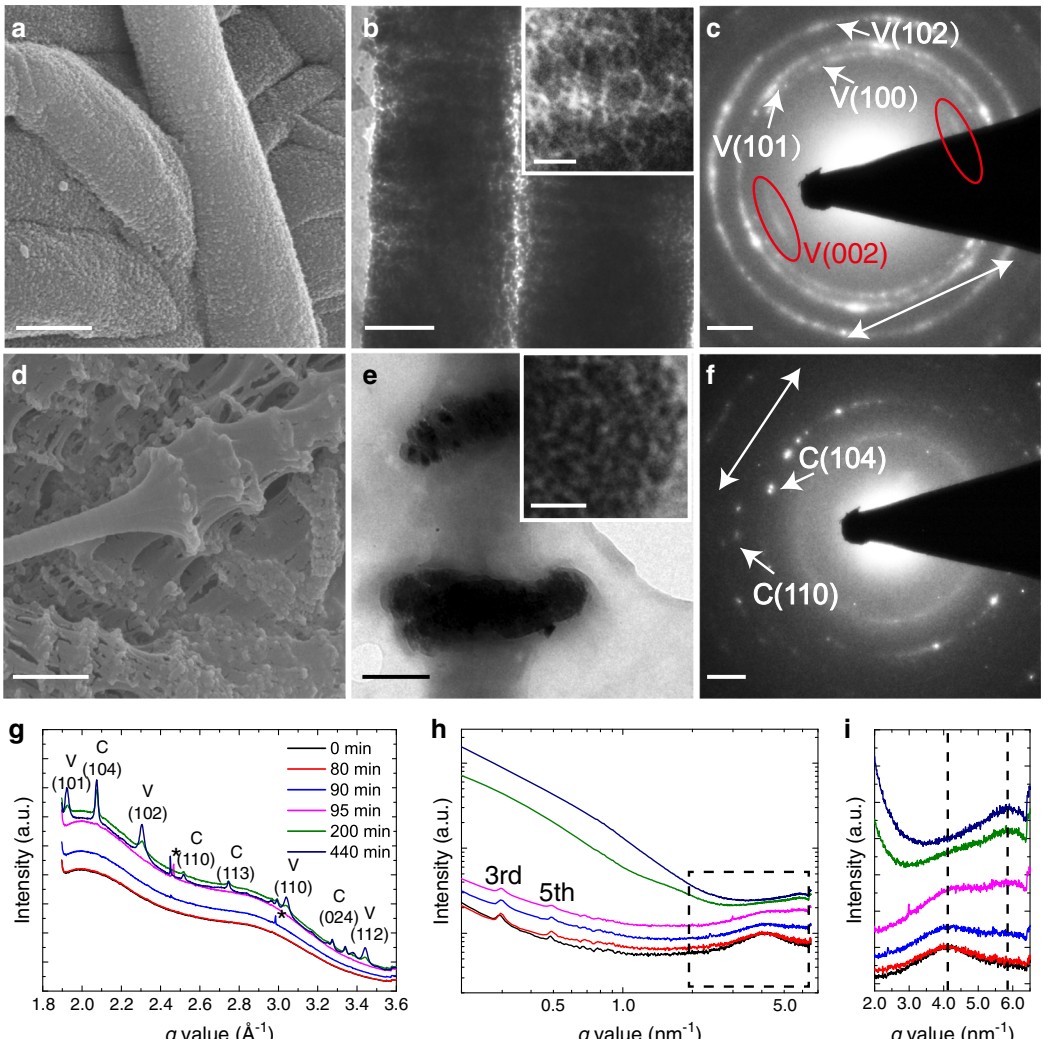

**Fig. 5 Type I collagen fibril mineralized with CaCO₃ in the presence of pAH. a, b** SEM and TEM images of collagen fibrils mineralized by vaterite after 48 h, respectively. Inset: higher magnification image showing the elongated morphology of the vaterite particles. **c** LDSAED of **b** showing arcs of vaterite (002) diffraction along the collagen fibril axis. **d, e** SEM and TEM images of collagen fibrils mineralized by calcite after 48 h, respectively. Zoom-in image in the inset of **e** shows the nanoparticle subunits. **f** LDSAED pattern of **e**. The direction of the collagen fibril long axes are highlighted by white arrows in **c** and **f**. **g, h** In situ WAXS and SAXS spectra of collagen fibrils during mineralization, respectively. * In **g** indicate the background signals from the X-ray or the mica windows. The 3rd and 5th order peaks corresponding to the 67 nm axial organization of collagen are highlighted in **h**. After long mineralization times, the signal from the CaCO₃ dominates, obscuring the signals of collagen. **i** Zoom-in of the square in **h**. The two vertical dash lines indicate the SAXS peaks related to the original lateral intermolecular distance of $d = 2\pi/4.08$ nm$^{-1}$ (~1.5 nm) and the compressed $d = 2\pi/5.68$ nm$^{-1}$ (~1.1 nm). Scale bars: **a, d** 500 nm, **b, e** 200 nm, Inset of **b, e**: 20 nm **c, f**: 1 nm$^{-1}$.

distribution (~±40°, Fig. 4e). In common with HAp, the γ-FeOOH platelets showed no preferred lateral orientation (Fig. 4d and f and Supplementary Movie 8).

The fibrils were also mineralized with calcium carbonate, which under ambient conditions can form three different polymorphs (vaterite, calcite, and aragonite)[46], that all exhibit crystal morphologies distinct from those of HAp and lepidocrocite. Mineralization was conducted by immersing type I collagen fibrils (bovine Achilles) in a solution of CaCl₂ and poly(allylamine hydrochloride) (pAH) and exposing them to ammonium carbonate vapor[47], where the pAH is a crystallization control agent[48], facilitating infiltration (Supplementary Fig. 15). The collagen fibrils became heavily mineralized within 48 h. The image contrast was insufficient for cryoTEM, so TEM was performed after freeze-drying vitrified samples on TEM grids.

Two types of crystals developed within the collagen fibrils. The first were ellipsoidal nanoparticles of dimensions 10 × 20 nm that

were elongated along the fibril axis (Fig. 5a and b). Higher densities of these crystals were present in the gap regions than in the overlap regions, resulting in a banding pattern (Fig. 5b). These were identified as vaterite using LDSAED, where the (002) reflections were present as a pair of arcs oriented along the fibril axis, while all other lattice planes appeared as rings (Fig. 5c). The second were ~100 × 600 nm calcite disks comprising ~10 nm nanoparticles. In common with previous observations[49] these were present in discrete regions in the collagen (Fig. 5d and e), and were neither elongated nor oriented in the collagen matrix (inset of Fig. 5e and f). Control experiments performed in the absence of collagen as expected[17], yielded micron-sized calcite and vaterite crystals with morphologies including thin films and fibers.

Scanning electron microscopy (SEM) demonstrated that CaCO₃ mineralization induced significant distortion of the fibrils (Fig. 5d). To understand the molecular basis of this distortion, we conducted in-situ analysis of the mineralization process using

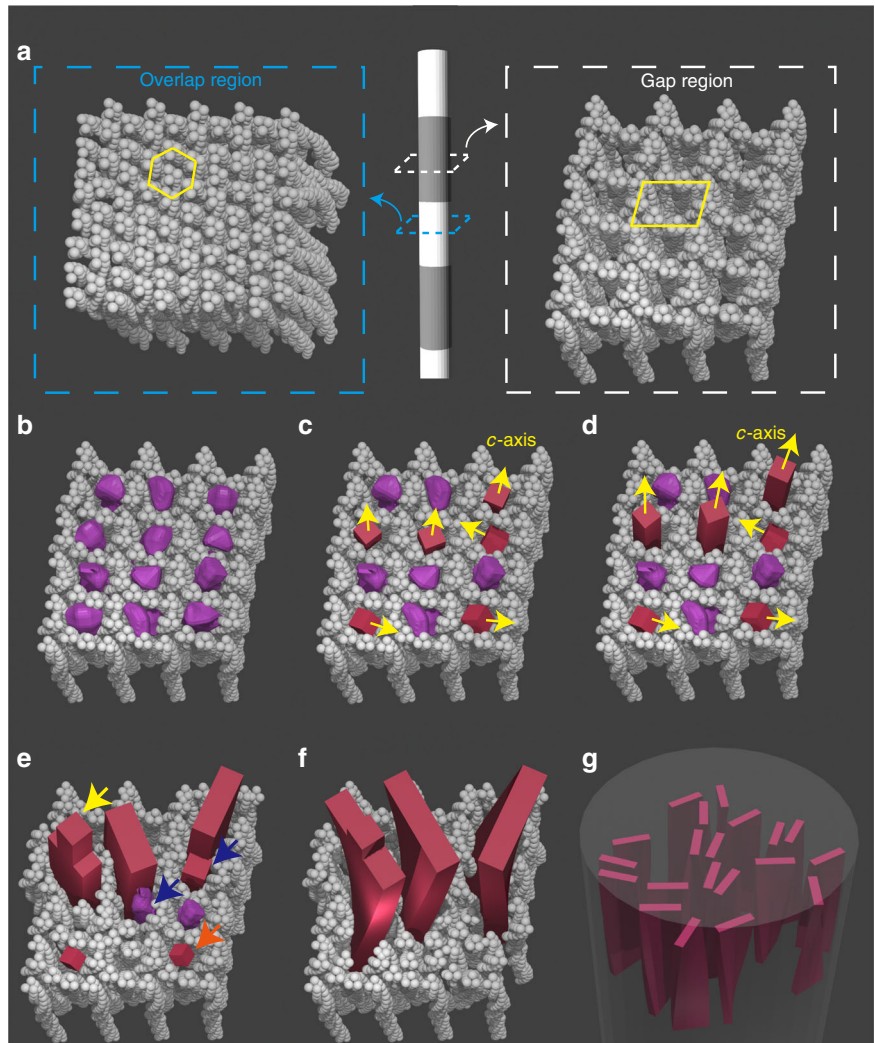

**Fig. 6 Schematic representation of the proposed confinement model for collagen mineralization.** The model is based on the X-ray crystal structure of the collagen fibril in Fig. 3, **a** where the overlap regions consist of tilted molecules which are densely packed in a quasi-hexagonal pattern (highlighted by yellow lines), and the gap regions contain straight channels (highlighted by yellows lines) allowing HAp nucleation. The channels are initially filled with ACP phase (purple particles in **b**), which through an amorphous-to-crystalline transformation form HAp nuclei (red particles in **c**). Nuclei with their fast growing *c*-axes parallel to the channels out-compete those with other orientations (**d**), and develop into platelets by pushing aside the collagen molecules, as shown in **e**. Needle-like tips are left on some platelets (highlighted by yellow arrow). Misoriented nuclei and ACP are either consumed by Ostwald ripening (highlighted by the orange arrow), or fuse with other platelets by adjusting the ionic arrangement[58] (highlighted by blue arrows). The platelets further grow and start to twist around their long axes probably due to the locally changing collagen organization (**f**). This results in the twisted HAp platelets being roughly uniaxially oriented with their *c*-axis parallel to the long axis of the fibril (**g**).

small-angle/wide-angle X-ray scattering (SAXS/WAXS) (Supplementary Fig. 16). No changes in the SAXS and WAXS spectra were observed during the first 80 min of the reaction, while an increase in the scattering intensity was detected after 90 min (Fig. 5g). This can be attributed to the formation of amorphous calcium carbonate (ACC) within the fibrils, where this causes a small decrease of the lateral packing distance of the collagen molecules. This is demonstrated by broadening of the SAXS peak at $q = 4.08$ nm$^{-1}$ (~1.5 nm), which corresponds to a reduction in the separation of the collagen molecules (Fig. 5h and i).

WAXS confirmed the development of both vaterite and calcite at incubation times >95 min (Fig. 5g), while SAXS revealed that crystallization of the ACC is accompanied by a significant reduction in the intermolecular distances of the collagen molecules from ~1.5 to 1.1 nm, as shown by the replacement of the $q = 4.08$ nm$^{-1}$ peak with one at $q = 5.68$ nm$^{-1}$ (Fig. 5h and i). The axial ~67 nm d-band organization remains unchanged

during this process. The amount of crystalline material in the sample then continued to develop with time and significant amounts of calcite and vaterite were observed after 440 min (Fig. 5g). Similar observations have been made for bovine[50], fish bone[51], and turkey tendon mineralized with HAp[52], demonstrating that the collagen molecules are pushed apart and compressed by initial infiltration of the ACC, and more significantly by its subsequent crystallization.

## Discussion

Historically, the first ideas about the structure of mineralized collagen came from the early TEM study of Weiner and Traub, who suggested that HAp platelets in mineralized collagen are co-aligned along all three crystallographic axes throughout a collagen fibril forming a deck-of-cards organization (Fig. 1d)[11]. This was later supported by the ET studies of Landis et al. [1,13]. A highly

ordered gap region structure was proposed to explain such an organization, in which the collagen molecules are 3D organized and form parallel 2D channels (Fig. 1a). In this well-defined scaffold, the amino acid residues and the distribution of charge in these channels are believed to induce HAp nucleation[9,21–23] and to epitaxially template the crystal orientation (Fig. 1b)[9,13,18,21]. While this scenario has never been confirmed experimentally, it has been widely accepted and is frequently described in textbooks and review papers (see Supplementary Note 1 for a list of influential papers presenting this model)[3,4,14]. Over the last decade, however, experimental and computational studies have questioned this model, both with respect to the orientation of the crystals in the collagen matrix, and to the way in which crystal orientation is achieved.

Burger et al. concluded from an X-ray scattering study of fish bone that although the HAp platelets were aligned with their c-axes along the collagen axis, the stacks of platelets had irregular intercrystalline spacings, and varied in their lateral orientations[19]. From STEM tomography Reznikov et al. proposed that the HAp platelets in human bone had random lateral orientations and only formed short stacks[2]. Both of these studies therefore support the conclusion of Olszta et al. that the platelets only exhibit uniaxial alignment[15].

Nevertheless, these studies do not provide a comprehensive picture for the structure of collagen and the way it is mineralized. Reznikov et al. did not discuss the origin of crystal orientation with respect to the collagen fibrils while Burger et al. proposed that parallel 2D channels exist within a collagen fibril (cf. Fig. 1a), which could template the formation of HAp platelet stacks. Deformation of the matrix during this process would also lead to distortion of the parallel alignment of the crystals[19]. Olstza et al. proposed that the HAp platelets form by transformation of the ACP confined in the space between collagen molecules, but did not yet have the details of how this intermolecular space was distributed throughout the collagen structure[15].

While the community had considered the collagen structure to consist of a quasi-hexagonally packed array of staggered, straight collagen molecules, in 2006 the XRD study of Orgel et al. revealed a more complex triclinic superstructure, in which the tropocollagen molecules are organized in a twisted and tilted arrangement[32]. One collagen microfibril comprises five 1D-staggered, twisted collagen triple helix molecules, where the staggering leads to the characteristic periodical band structure that contains a dense overlap region and a more loosely packed gap region[32]. Under this structural model, which was confirmed by a recent computational study of Xu et al.[36], a hole zone with 2D channels as depicted in Fig. 1a does not exist.

Looking then from the perspective of the mineral phase, Wang et al. used solid-state nuclear magnetic resonance (ssNMR) to reveal that the HAp platelets—even in mature bone—are covered by a hydrated amorphous layer[24,31], which prevents any direct molecular recognition between collagen and HAp. Nevertheless, some recent studies have suggested that the charged amino acids in collagen are able to nucleate calcium phosphate minerals[22,23,30], and that an epitaxial match (cf. Fig. 1b) exist with calcium-deficient apatite[18]. More consistent with our new insights on collagen-directed mineralization is the proposal that the orientation of HAp crystals is simply induced by confinement within the gap region[15,19]. Indeed, precipitation of HAp in various in vitro systems including nanotubes[25], peptide amphiphile nanofibers[53], polymerized liquid crystals[54], and cellulose fibers[55] results in crystals whose c-axes are aligned with the long axis of the confining media.

The work we present here comprehensively addresses all three key aspects of the mineralization of collagen: (1) the organization of intrafibrillar HAp crystals, (2) the structure of the collagen gap region, and (3) the interaction between the mineral and the organic collagenous matrix. Our high resolution/high contrast TEM study visualized both the mineral and the collagen fibrils in bone and unequivocally confirms that the intrafibrillar HAp platelets are only uniaxially aligned with respect to the collagen long axis[15], and that they sometimes form small stacks of platelets[2]. Notably, these data were obtained from the analysis of single collagen fibrils rather than bulk bone[2], or bundles of collagen fibrils[19], such that the structural information is unambiguously related to intrafibrillar crystals only.

We then demonstrate that the gap regions—where the HAp crystals nucleate[12]—contain discrete, elongated channels that start in the a-band region. Based on our previous observations of an in vitro model system, which show that mineral infiltration in the form of ACP begins in these sites[38], we propose that the first crystals form by the transformation of the ACP in the channels, resulting in the formation of needle-like crystals whose sizes and shapes are defined by the geometry of the channels. Our observations therefore necessitate that the traditional model based on 2D gap zones is revised, and provide the foundation for a model of collagen mineralization based on confinement (Fig. 6) rather than on specific interactions between the mineral and organic matrix components. This model is further supported by our demonstration that collagen also induces the crystallographic and morphological alignment of γ-FeOOH and vaterite, in a manner comparable to that of HAp.

In line with our model, Traub et al. have shown that the intrafibrillar HAp crystallization starts in the gap regions within the collagen fibrils[12]. Additionally, it has been shown both in vivo[28] and in vitro[38] that collagen mineralization proceeds through an ACP precursor that subsequently transforms into crystalline HAp. We therefore propose that ACP readily fills the channels in the gap regions (Fig. 6b), and that smaller amounts of ACP will infiltrate into the tightly packed overlap regions where intermolecular spacings are <1 nm (Fig. 6a). This is likely to be a synergistic process where the collagen defines the shape of the ACP, and the ACP infiltration induces some distortion of the collagen structure, as is also observed for $CaCO_3$ (Fig. 5g–i). A population of randomly oriented HAp nuclei will then form in the channels, either via dissolution-recrystallization, or pseudomorphic transformation of the ACP precursor (Fig. 6c)[56].

Based on previous in vitro experiments using cylindrical nanopores[25] we expect that only those crystals oriented with their fast growing c-axes aligned with the channel (and thus the fibril) axis are able to grow unrestricted into needle-shaped crystals (Fig. 6d). Crystals that are oriented in other directions can only grow by pushing apart the collagen molecules, which is energetically less favorable. Those misoriented smaller crystals and remaining ACP will subsequently be consumed by the growth of the needles, either by a dissolution-reprecipitation process (Ostwald ripening)[57], or by adjusting the ionic arrangement and fusing with the growing HAp crystal[58]. The lateral growth of the needles will then generate platelets by pushing aside neighboring collagen molecules (Fig. 6e).

This mechanism is possible due to the unique structure of collagen, where the gap region is more compliant and compressible than the overlap region[59]. Continued growth of the crystals along the c-axis will cause them to extend into the overlap region, causing a rearrangement of the collagen molecules, and a concomitant displacement of the included ACP. This is supported by our SAXS/WAXS data on $CaCO_3$ (Fig. 5g–i) that show the rearrangement of the collagen during the amorphous to crystalline transition. During this process, the platelets also start to twist along their long axes in agreement with previous reports (Fig. 6f)[2], which we tentatively attribute to the locally changing collagen organization within the microfibril. This eventually leads to

twisted HAp platelets that are uniaxially oriented along their *c*-axes, where adjacent platelets may direct each other and form small stacks, depending on the degree of mineralization (Fig. 6g, see also Fig. 2f and n).

This scenario is in excellent agreement with studies of fish bones[19] and human bones[2], which showed that HAp nanoplatelets[2,12] only form short or irregular stacks, that may become intergrown in time[2]. Furthermore, the recent study of Reznikov et al. also observed needle-like tips on HAp platelets in human bone that resemble fingers of a hand as well as individual needle-like crystals[2], and proposed that the HAp nanoplatelets develop from these needle-like crystals. We note that a similar scenario was suggested as far back as 1992 by Traub et al. based on the observation of needle-like crystals in the early stages of turkey tendon mineralization[12]. We emphasize that the final intrafibrillar mineral principally comprises crystalline HAp nanoplatelets, rather than needle-like crystals[60] or ACP[61] as suggested by some dark-field TEM (DFTEM) studies. These discrepancies may be related to the fact that only a small fraction of the total diffraction signals were used to form DFTEM images. As the intrafibrillar HAp nanoplatelets have an imperfect uniaxial orientation and can also be twisted, most will have no contrast or only be partly visible in the DFTEM images, making it difficult to deduce their crystallinity and morphology with this technique.

γ-FeOOH and vaterite also have anisotropic crystal structures and grow fastest along their *c*-axes[45,62], resulting in a preferred alignment of this axis with the collagen fibril. Calcite, in contrast, has a rhombohedral crystal structure and no preferential growth direction[63], such that no orientation was observed within the collagen fibrils. That the degrees of orientation of γ-FeOOH and vaterite are less pronounced than for HAp can be attributed to the smaller aspect ratios of the intrafibrillar γ-FeOOH and vaterite crystals (1.9 and 2.0, respectively) as compared with HAp (4.1). The gap zone channels therefore have a weaker influence on their crystallographic orientation.

Although collagen can effectively direct crystal orientation, it has a relatively small effect on crystal morphologies. The HAp and γ-FeOOH crystals formed within the collagen have nanoplatelet morphologies comparable to those formed in bulk solution, and are just 2–3 times smaller in length and width. The intrafibrillar vaterite nanoparticles also closely resemble the subunits from which polycrystalline vaterite particles are usually constructed. In all of these cases, the width of the final crystals exceeds the original width of the channel in which they form. This shows that the collagen matrix is flexible enough for a crystal to push the collagen molecules aside, accessing a neighboring pore to develop into its preferred morphology (Fig. 6c). In contrast, calcite—which invariably appears as micron-sized, rhombohedral crystals in bulk solutions—grows to large sizes, vastly distorting the fibrils. These observations suggest that the final morphologies of the intrafibrillar crystals are largely determined by their crystallographic structure rather than by the collagen matrix.

While the oriented growth of intrafibrillar HAp in bone can be explained based on principles of physical chemistry alone, we also emphasize the importance of biological control mechanisms in the formation of this complex hierarchical tissue. While our model shows that the uniaxial orientation of HAp in collagen fibrils can be induced by confinement only, it does not exclude the involvement of organic matrix components in the nucleation process[22,23,64], such as the amino acid groups lining the channels, carbohydrate modifications of collagen, or osteocalcin entrapped in the gap region. In particular, one could envision a role for charged groups in creating a $Ca^{2+}$ sponge to increase the local supersaturation, promoting nucleation in the gap region. This has been proposed, not only for calcium phosphate[12], but also for the nucleation of $CaCO_3$ in the nacre of mussel[65,66].

It is also recognized that additional soluble molecules including citrate[67] and many non-collagenous proteins are involved in collagen mineralization[68]. Here, we emphasize that in contrast to most other reports, the data from our study can be unequivocally attributed to intrafibrillar mineralization. Therefore, these non-collagenous molecules are unlikely to be involved in the orientational control of intrafibrillar crystal formation. However, they most certainly play a role in intrafibrillar mineralization by facilitating mineral infiltration[69], as has been demonstrated for polyelectrolytes (e.g., pAsp) in in vitro model systems[15,38]. These molecules may also be involved in the control of extrafibrillar mineralization, which accounts for a significant fraction of the mineral content in bone[70].

Finally, our model does not address the higher levels of hierarchical organization of the mineral platelets[2,3,69] that are responsible for the 3D mechanical anisotropy of bone[70,71], nor does it describe the rotated-plywood organization of the crystals due to the twisting of collagen fibrils[70], or the formation of extrafibrillar minerals[2,8]. Again, multiple bio-molecules are inevitably involved in these complex processes.

The exceptional properties of bone are intimately linked to its unique hierarchical structure, of which the mineralized collagen fibril is the basic building block. Identification of the structure of the mineralized collagen fibril, and the mechanisms that underlie its formation, therefore underpins the development of new strategies for promoting bone regeneration and creating biomaterials with properties comparable to those of bone[54,55]. The long-standing deck-of-cards model of intrafibrillar HAp, and the belief that orientation is achieved via an epitaxial chemical interaction between the organic matrix and nascent crystals, have made it extremely difficult to replicate this structure. We demonstrate here that the reality is actually considerably less complex: the HAp platelets are only uniaxially oriented, and that this is achieved via generic confinement effects. This minimalistic approach to the formation of highly ordered assemblies by crystallization from solution is also known to operate in other biological systems, where biomolecules are only organized to create the boundaries for physical chemistry to do its work[72,73]. We anticipate that application of similar simple strategies will facilitate the design of synthetic systems that mimic bone or promote its regeneration[54,55].

## Methods

**Materials**. Analytical grade $CaCl_2$, $K_2HPO_4$, $FeCl_2$, $FeCl_3$, $(NH_4)OH$ aqueous solution (28%), $(NH_4)_2CO_3$, 4-(2-hydroxyethyl)piperazine-1-ethanesulfonic acid (HEPES), uranyl acetate, poly-(α, β)-DL-aspartic acid sodium salt (pAsp, molecular weight = 2000–11,000 g mol$^{-1}$) and poly(allylamine hydrochloride) (pAH, mw = 15,000 g mol$^{-1}$) were purchased from Sigma-Aldrich and used without further purification. Type-I collagen extracted from horse tendon was kindly provided by Prof. Giuseppe Falini (Department of Chemistry, University of Bologna, Italy) and was originally purchased from OPOCRIN Spa[38]. Type-I collagen sponge tapes derived from bovine Achilles tendon was purchased from ACE Surgical.

**Preparation of TEM lamella of bone sample**. The present study was carried out on a sample of healthy cortical bone from the left tibia of a 10-year-old female. According to the Central Committee on Research involving Human Subjects (CCMO), this type of study does not require approval from an ethics committee in the Netherlands (see https://english.ccmo.nl/investigators/legal-framework-for-medical-scientific-research/your-research-is-it-subject-to-the-wmo-or-not). A section of ~2 cm × 2 cm × 2 cm, containing cortical bone was cut and pre-fixed in a solution of 2% paraformaldehyde (PFA) in cacodylate buffer at pH = 7 for 2 days. After pre-fixation the samples were washed using ultrapure water on a rocking table (for two periods of 6 h and subsequently overnight) which were fixed again with 4% glutaraldehyde in cacodylate buffer at pH = 7 and washed with double-distilled water in steps as above mentioned[74].

Later, fixed samples were stained with the OTOTO procedure (also known as conductive staining)[75], involving the sequential exposure to osmium tetroxide (O) and thiocarbohydrazide (T) providing contrast as well as conductive properties[2,3]. After the staining procedure, bone pieces were washed with double-distilled water for 2 min and embedded in Epon epoxy resin dissolved in acetone (Embed 812, EMS, USA). The Epon embedding was carried out in five steps, each for 2 h: 25%, 50%, 75%, 100% repeated twice, which was followed by 100% resin overnight and

final embedding in a mold for 48 h at 60 °C. After 48 h, Epon-embedded bone pieces were trimmed using a glass knife on a microtome order to expose the embedded tissue. The trimmed blocks were mounted on a house made sample holder using silver paste and sputter-coated with gold. The TEM lamella preparation was performed in a dual-beam FIB/SEM (FEI Quanta FEG 600, Thermo Fisher Scientific), equipped with a gallium ion source operating with an accelerating voltage range of 0.5–30 kV and an Omniprobe™ micromanipulator. We prepared a 100 nm-thick section from the bone sample for transmission electron microscopy studies using the FIB lift-out technique. The region of interest for TEM lamella was determined after visualizing the surface of bone pieces by SEM. A thin section was cut parallel to the orientation of the long axis of the bone meaning that collagen fibrils would have in-plane view upon visualizing the thin section by TEM. The thin section ($10 \times 10 \times 0.1$ μm) was transferred to a three post lift-out TEM half-grid (Agar Scientific AGJ420) in order to use it for further ET characterization.

**Analysis of collagen structure**. The electron density map of $8 \times 8 \times 1$ collagen unit cells was built using the UCSF Chimera package[76], based on the structure obtained by a previous X-ray study[32]. The map was generated with a resolution of 0.3 nm and voxel size of 0.1 nm. From this electron density map we segmented the intermolecular channels within the collagen structure by applying in-house Matlab scripts. The edges of the channels were defined by considering the diameter of each collagen triple-helix molecule as 1.5 nm[77]. Channels with diameters <8 pixels (0.8 nm) were removed from the segmentation by eroding and then dilating the segmented channels for 4 pixels for an enhanced separation of the main channels. The channels on the model surface were also removed for improved visualization of the channels inside the collagen structure. The remaining channels were then labeled with different colors according to their connectivity, and visualized in 3D using Amira-Avizo software (Thermo Scientific™).

**Mineralization with HAp**. Horse tendon collagen fibrils (purchased from OPO-CRIN Spa) were prepared and mineralized with HAp[38]. These collagen fibrils were self-assembled on cryoTEM grids and incubated in HEPES buffer (10 mM, pH 7.4) containing $CaCl_2$ (2.7 mM), $K_2HPO_4$ (1.35 mM) and pAsp (10 μg mL$^{-1}$) at 37 °C for 72 h. After mineralization, the grids were removed from the mineralization solution and subsequently washed with MilliQ water, incubated in 0.5% uranyl acetate in MilliQ water for 15 s. After that the grids were washed with MilliQ water for 1 min, manually blotted and vitrified using an automated vitrification robot. For the experiments using bovine Achilles collagen fibrils, the fibrils were separated from collagen sponges (purchased from ACE Surgical) by thoroughly grinding the sponges into a fine powder in liquid nitrogen and drying at 37 °C. The powder was then dispersed at a concentration of 5 g L$^{-1}$ into Hepes buffer (10 mM, pH 7.4) containing $CaCl_2$ (4.5 mM), $K_2HPO_4$ (2.1 mM), and pAsp (75 μg mL$^{-1}$) and heated at 37 °C for 4 days[15]. After mineralization, 3 μL of solution was applied to a cryoTEM grid and vitrified.

**Mineralization with lepidocrocite**. Mineralization of collagen fibrils with lepi-docrocite was performed by incubating 0.125 cm$^3$ of collagen sponge (bovine Achilles, ACE Surgical) in a solution of $FeCl_3$ (1 mM), $FeCl_2$ (0.5 mM), and pAsp (0.69 mg mL$^{-1}$, Asp residue:Fe ions = 4:1) overnight. The solution was subse-quently placed in a glove box under $N_2$ saturated with 8% $NH_4OH$ for 24 h. The increase in pH was constantly monitored using a TIAMO Titrando Set-up (Metrohm). Control experiments were performed with different concentrations of pAsp. After the reaction, the sponge was washed thoroughly with ethanol, and ground in liquid nitrogen for TEM analysis.

**Mineralization with calcium carbonate**. To mineralize collagen fibrils with cal-cium carbonate, collagen sponge (bovine Achilles, ACE Surgical) was ground in liquid nitrogen as described above, and dispersed in solutions of 10 mM $CaCl_2$ at a concentration of 5 g L$^{-1}$, in the absence of further additives or in the presence of 1 mg mL$^{-1}$ pAH. The solutions were then transferred to a desiccator in which they were exposed to the $CO_2/NH_3$ gaseous mixture released on the decomposition of solid $(NH_4)_2CO_3$. 3 μL of the reaction solutions were collected at different time points and vitrified. For SEM studies, the samples were added to ethanol, cen-trifuged ($4715 \times g$) and dried at room temperature. The in situ X-ray studies were performed at the DUBBLE beamline at the European Synchrotron Research Facility (ESRF), Grenoble, France, by incubating 0.125 cm$^3$ of collagen sponge (bovine Achilles) in a flow cell between two mica windows (Supplementary Fig. 16)[78].

**Electron microscopy analysis**. SEM and energy-dispersive X-ray spectroscopy (EDX) studies were performed using an FEI Quanta 3D field emission SEM equipped with an EDAX EDX detector. Au quantifoil grids were used for cryoTEM sample preparations, and the vitrification was performed using an automated vitrification robot (FEI Vitrobot™ Mark III). (Cryo)TEM imaging was typically performed under ~5 μm defocus on a FEI-Titan TEM equipped with a field emission gun and operating at 300 kV. Images were recorded using a 2k × 2k Gatan CCD camera equipped with a post-column Gatan energy filter (GIF), with an electron dose of lower than 10 e$^-$ Å$^{-2}$ per image. For bone samples and collagen

fibrils mineralized with lepidocrocite, the tomography tilt series were taken by tilting the specimen from approximately −65° to 65°, 2° per step. For collagen fibril mineralized in vitro with HAp, the tomography tilt series was taken by tilting the specimen from approximately −60° to 60°, with a low-angle tilting step of 2° (from −45° to 45°) and a high-angle tilting step of 1° (<−45° or >45°). The electron dose of all the tomography tilt series is 1.0 e$^-$ Å$^{-2}$ per frame. The alignment and 3-dimensional reconstructions of the data sets were performed in IMOD. Image analysis was carried out using Gatan Digital Micrograph and Matlab.

## Data availability

All of the data that support the findings of this study are available online at: Supporting Data for "Intermolecular Channels Direct Crystal Orientation in Mineralized Collagen", Figshare, https://doi.org/10.6084/m9.figshare.12845990.

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

## Acknowledgements

We thank Dr. Koen Pieterse (Institute for Complex Molecular Systems, Eindhoven University of Technology, the Netherlands (TU/e)) for his help with data analysis, Zaf Khalil (Department of Chemical Engineering and Chemistry, TU/e) for his help with bone sample preparation and Andreas J. Fijneman (Department of Chemical Engineering and Chemistry, TU/e) for helpful discussions. We thank Dr. Elena Macias (Radboud Institute for Molecular Life Sciences) for her critical review of the manuscript. This work was

supported by the Netherlands Organization for Scientific Research (NWO) through VICI (F.N., M.J.M.W.) and Toppunt (Y.X.) grants to N.S. E.D.E. and N.S. were supported by the European Research Council (ERC) Advanced Investigator grant (H2020-ERC-2017-ADV-788982-COLMIN). Y.X. was supported by the Marie Curie Individual Fellowship (H2020-MSAC-IF-2019- 885795-PolyTEM), and A.A. was supported by the Marie Curie Individual Fellowship (H2020-MSCA-IF-2017-794296-SUPERMIN). E.D.E. and A.A. were also partially supported by an NWO Echo Grant. W.B.'s contribution is partially based upon work supported by Oak Ridge National Laboratory, managed by UT-Battelle, LLC, for the U.S. Department of Energy. The work was further supported by an Engineering and Physical Sciences (EPSRC) Platform Grant to F.C.M. (EP/N002423/1), an EPSRC Program Grant (EP/R018820/1) which funds the Crystallization in the Real World Consortium (F.C.M.), and a Leverhulme research project grant (F.C.M., N.S., and Y.X.). NWO is also gratefully acknowledged for making access to the DUBBLE ESRF beamline possible. This project was also supported by grant 9 P41 GM103622 from the National Institute of General Medical Sciences of the National Institutes of Health. We also thank the University of Edinburgh for financial support.

## Author contributions

F.C.M. and N.S. originated and co-supervised this project. W.H.N., E.D.E., A.A., and Y.X. prepared and analyzed the bone sample. Y.X., C.O., and H.F. performed the analysis of collagen structure. Y.X. and F.N. performed the collagen mineralization experiments and TEM study. E.D.E., M.J.M.W., F.N., and Y.X. performed analysis of TEM data. B.C. helped with CaCO$_3$ mineralization experiments and together with F.N., D.H.-M., G.P., and W.B. performed the in situ X-ray study. P.H.H.B. provided support on TEM studies. H.F. supervised TEM data analysis. J.P.R.O.O. supervised the analysis of collagen structure. A.A. supervised bone sample preparation. Y.X., F.C.M., and N.S. co-wrote the manuscript. All the authors have read and commented on the manuscript.

## Competing interests

The authors declare no competing interests.
