## [Peer Review File · Nature Communications]

Reviewers' Comments:

Reviewer #1:

Remarks to the Author:

Review on "Intermolecular Channels Direct Crystal Orientation in Mineralized Collagen" by Y.F. Xu et al.

The authors describe a mechanistic model for HAP plates generation in bone. They state that 2 nm channels in the gap region of collagen serve as template for initial HAP needle seeds oriented [001] along the channel. Further, lateral growth of the needle induces a plate formation by keeping c-axis orientation parallel to collagen fibril long axis.

Chemical templating or involvement of non-collagenous proteins is not needed for the explanation of the bone composite material formation. Further on, they use γ -FeOOH and CaCO₃ to show the general principle of their model.

General remarks:

The work is excellent presenting a lot of investigations including X-rays, cryo TEM on bone and on a model system comprising in vitro mineralized collagen.

Detailed remarks:

- Length of channels known?

- Is it possible to have different HAP needles along the channels, and are they single crystals?

- Did the in vitro-mineralized collagen aspartic acid serve as non-collagenous inhibitor? How does the low pH of the acid influence the mineralization process compared to mineralization without?

Formation of acidic CaP (Brushite, octacalcium phosphate) preferred?

- Did the authors observe other calcium phosphate phases such as tricalcium phosphate, brushite or octacalcium phosphate (for literature see e.g. Vyalikh et al. Early Stages of Biomineral Formation—A Solid-State NMR Investigation of the Mandibles of Minipigs. Magnetochemistry 2017, 3, 39; doi:10.3390/magnetochemistry3040039)?

In the electron diffraction, around the primary beam the small angle reflections are over exposed thus the observation of these phases remains difficult e.g. the 100 reflection of octacalcium phosphate. On the other hand, it is clear that needle shape crystals should be HAP, but at initial state also meta stable phases could be formed (OCP e.g. in young bone and dentine).

- Why do not appear the initial needle at all plates? Is it etched away or dissolved during the growth procedure or is it simple too small to be recognized in the final HAP plate?

HAP plates can have also roundish edges, thus the initial needle should form a straight linear edge and thus identified probably in the plate?

- If the mineralization starts in the gap region, does it mean that the overlap region remains unmineralized?

- Did the authors tried to isolate the needles from the collagen by dissolving it at the initial state of mineralization? In this way they would have a direct prove for their model in Fig. 6b and show in HRTEM individual HAP needles as prove!

- Besides HAP and others also the possibility of the formation of amorphous calcium phosphate has to be considered as discussed e.g. in an actual paper by Swarcz et al. in Bone 153, (2020) 115 304. They also discuss the topic which is widely held in theory that the gap zone should reside the most of the mineral in bone. Can the authors exclude formation of amorphous calcium phosphate in the channels at least in the early stage of mineralization? Of course, it is difficult to prove by electron diffraction since it gives no signal (reflections) in the ED patterns. Thus, applying of

HAADF or EELS/EDS could only help in TEM. The presence of (suprafacial) amorphous calcium phosphate on HAP platelets was shown with NMR by von Euw (Bone mineral: new insights into its chemical composition. Scientific Reports (2019) DOI: 10.1038/s41598-019-44620-6)

Conclusion:

For me the work seems to be conclusive and nice, nevertheless it is in contradiction of classical view that chemical epitaxy or non-collagenous proteins should also determine bone mineralization. Thus, I recommend this paper for publications in Nature Communications, after answering to the above questions

Reviewer #2:

Remarks to the Author:

General Comments:

This paper is interesting and well written with only a few minor errors (see detailed comments), and I agree with the main conclusions that this data suggests that no specific interactions with biological macromolecules are required to give similar crystal faces and orientations as bone, and agree that much of the structure can be reproduced by general physical chemistry. Of course I agree because I have been making that same argument for years. This was already demonstrated in the Olszta paper several years ago that developed the first model system. Then the Cantaart/Meldrum paper showed quite definitively that hydroxyapatite grown under confinement will inevitably lead to the same [001] orientation as bone. So to me, no revolutionary new concepts have come from this work. Nevertheless, this paper elegantly demonstrates the concepts further by examining two other, non-bone mineral systems. I like this idea, and in fact had come up with the same idea several years ago and submitted an NSF proposal to do exactly the same thing, but I had no success in getting it funded. So I'm glad the authors did it here, and especially since they are so much better equipped with the microscopy instruments and skills. Although I think Meldrum already hit the nail in the coffin with the Cantaart paper by demonstrating that HAp will obligingly form oriented crystals with sufficient confinement, it seems other groups keep arguing for protein epitaxy relationships no matter what. So maybe this paper with its lovely images and dual systems will help overcome that unrelenting belief, so that no more grant money has to be wasted. So for this reason, even though I didn't learn that much in this paper, I think others might. Having said that, I have several suggestions which I think will make this paper more impactful.

This paper nicely models the channels within a collagen fibril, but I wouldn't say they have proven that, as the title of the paper implies. In contrast, the recent Reznikov and Kroger paper seems to demonstrate that the crystals meander in and out of the neighboring collagen fibrils, so there seems to be a disconnect between your hypothesis and their observations of bone. Just showing a model of extra space in the gap zone doesn't actually prove that the crystals are forming within any predefined channels, especially considering the crystals grow beyond those supposed channels into the overlap zones. Having said that, I think you might actually data that does show some evidence of channels in your video movie S5. In this movie of the isolated model system fibril, there are some interesting patterns if one slowly moves the frames back and forth as you focus on the dots at the outer edge of the fibril. I can see "streams" of black dots that appear to extrafibrillar, and then they often take a sharp curve inward as they infiltrate into the fibrils. I suspect these dotted streaks are still actually amorphous nanoparticles (or PILP-like nanodroplets) that seem to line up prior to entry into the fibril (as was previously shown in your Nat. Mat. 2010 paper). But in this movie one can see that the connected dots seem continue along a connected path that infiltrates into the fibril, appearing to be a channel. This data might even support the capillary infiltration mechanism. However, I understand if you don't want to go into infiltration mechanisms here since this paper is focused on orientational growth inside the fibril. But it does nicely illustrate the interesting channels of entry, which could perhaps be commented on without trying to pronounce any particular mechanism, yet supporting the argument that there are some

type of channels. Whether those channels are pre-existing or not may remain to be determined. But I assume you could determine where these channels are starting along the fibril gap zone or specific band as you did in your prior paper, and a consistent starting location might suggest a consistent entry channel position. This might also fill the gap between the Reznikov observations and your model system, because it shows a pre-crystal that spans the outside and inside of a fibril, all connected by one dotted path.

I think your section starting at line 344 on the confinement-based model is inconsistent with your prior work, as well as bone studies, because you seem to be describing crystal growth as entering into the overlap zone, when your prior work, as well others, shows there is amorphous precursor throughout those regions as well, prior to the streaks of crystals. So the confinement is really of the ACP precursor phase, and not crystals pushing their way into the denser collagen region, right? Crystals growing from within an amorphous phase is also a form of confinement, so if the ACP is confined first, then the crystals will likely be as well. Especially if it follows roughly a pseudomorphic transformation (although dissolution and recrystallization can't be ruled out). In any case, I think you need to rework that section in remembering there is an ACP phase present that is initially confined.

The final schematic (Fig. 6) is nice for representing the random lateral orientation of the crystals, but one can see that in the actual images anyway. The use of a schematic would be better for illustrating the orientation mechanism, which is the crux of your story, and one which the community doesn't seem to have grasped yet. It should show how the crystal nuclei presumably start off more random, but then become oriented as the rapid growth direction takes over. In addition, you show the same rectangular platelets as the old literature, but a key part of your message is that they twist as they traverse down the fibril; so I think your schematic should show that as well. And relating to my prior comment, I think the interstices should show an amorphous phase from which the crystals nucleate and grow, rather than crystal tips growing into the space. It seems you are focused on the argument that it is these crystal tips that have fooled people into thinking the crystals are needles, but this part of your story is very weak. I'm pretty sure this is not the cause of the needle-like observations by many others, and certainly not in the recent report by Reznikov who tracked the crystal trajectories and did not make the mistake of thinking the tips of a growing platelet was a needle. They argue that there is fusion of the originally needle-like crystals. Once again, you do have data that could better illustrate that, because the movie S5 shows a more needle-like morphology. This is presumably early stage mineralization since the crystals streaks are quite dotted, and apparently the nanoparticles/droplets are not fully fused and/or crystalline. So in my opinion, I think this section could be reworked, and you could do a better job in this schematic of illustrating how the crystals become uniaxially oriented but twist, as well as illustrating what is actually confined (the ACP phase), and how the ACP phase and early stage crystals infiltrate and are more needle-like.

Detailed Comments:

- I love the videos of the fibril cross-sections which nicely illustrate the propeller like orientational rotation of the crystals along the length. The explanation and demonstration of why the old Landis reports only saw biaxial oriented crystals was nice, and in that case was respectful.

- Pg 3, line 75. "it has been suggested that the orientation of the crystals may be governed by their confinement within the gap zone of the collagen.^{15,18} However, for such a mechanism to operate the gap zones could not contain the flat, 2D channels present within the traditional model of collagen, as these would in fact allow HAp to grow with its c-axis in any direction parallel to the channels, rather than preferentially along the long axis of the fibrils.¹⁷" The Olszta paper did not claim the crystals were confined only within the gap zone. Quite the opposite. Others suggested the crystals may nucleate in the gap zone, but our argument was that the precursor might infiltrate there, but the crystals clearly could outgrow the gap zone and we proposed the rapid growth direction of HAp leads to orientation. Which is exactly what you are arguing for here. I think you are just trying too hard to prove you have come up with something new, but it's mostly not, just much more nicely illustrated.

- pg 4, line 93: "This provides compelling evidence that the orientation of HAp within collagen fibrils arises due to confinement effects only." I think the compelling evidence was already provided by the Cantaart model system, which shows the HAp will show preferred orientation in constrained nanopores of non-specific interactions, period! That should have been the end of this debate.
- Fig. S2. I find your wording on the captions to be a little confusing. I would have thought a "top view" of the fibril would be like a top down view (looking down the c-axis). Seems like these are side views of the longitudinal axis of the fibril. I guess you are referring to top view of stacks, but some clarification might be useful since not everyone is well versed with tomographic techniques.
- pg. 6, line 129: "and usually propagated into the fibrils (Supporting Information 3)." This data is not convincing unless one knows for sure those are two different fibrils; I'm not sure how that was known other than the color overlay you provided. Not that I doubt that occurs, given the platelet tracking done in the Reznikov paper. But this statement might raise some brows.
- Fig. 3: It might be nice to put markers of where the e1 and d bands on the schematic since they are referred to in the text. (and maybe channel entry points as mentioned above).
- pg. 8, line 171: "and are markedly different from the traditional picture which depicts 2D channels that run parallel to the fibril axis (Figure 1).1,13,18,30" I don't think anyone ever thought that only a rectangular channel is formed in 2D. We all know there must be longitudinal channels, but it is just difficult to draw in a schematic. Now we know they are slightly tilted, so even harder to draw. Even your collagen in Figure 3 shows something that looks like a 2D rectangular region of channels. Likewise, schematic in Fig. 6 doesn't show longitudinal channels either beyond the gap zone, so I don't see why you are saying your perspective is markedly different.
- pg 6, line 146. I think comparing the tip of a platelet as being related to literature describing needles is not something to be correlated. Those reports were usually looking down the length, and not at the tip of a platelet. It is more likely the old edge-on view, and/or early stage mineral.
- pg. 7, line 152. "both of which limit the detection of HAp platelets that are not oriented parallel to the electron beam." I think you mean the platelets oriented edge-on were needed for contrast?
- pg. 8, you describe various dimensions and orientations in the in vitro model system as being similar to those in bone, but the images and videos have such a different appearance. Such as why do the regions next to the crystallites in bone appear brighter, almost a shadowing appearance, which isn't seen at all in the model system? Also, why are the in vitro crystallites seen as dark speckled platelets rather than solid streaks as in bone? Is it because it is not a slice in a dense sample; the staining that was used differs; the imaging method of stacked slices? Are the speckles because they are still amorphous nanoparticles in the early stage? I'm not doubting the data, just wish I understood the differences in imaging techniques and/or samples.
- Fig. 4 caption: "were measured that were at least 10 nm apart for (g) and 50 nm apart for (g)." There is no (g) or (h) images in figure 4.
- pg. 31, line 770: typo "magnification mage showing"
- pg 13, line 299: "Reznikov et al. did not discuss the origin of crystal orientation with respect to the collagen fibrils, while the models presented by Olszta et al. and Burger et al. were still based on the hypothesis that collagen fibrils contain 2D channels (cf. Figure 1a)." I don't understand why you are saying this. Reznikov tracked the collagen and crystal orientation, and showed the crystals exited and entered fibrils, so a pre-existing channel that you are trying desperately to prove may not exist. And your collagen model shows the gap zone contains short channels that stacked together look quite similar to the supposed 2D channels represented in other models. Plus, you are misrepresenting the Olszta paper, given that their schematic in Fig. 5 does NOT show a deck-of-cards arrangement of the crystals, nor does it show crystals only forming in the hole zones. It was deliberately arguing against those things in the text. While I appreciate being cited for uniaxial aspects earlier, in this sentence it gives the false impression that our paper talked about 2D channels and argued for biaxial orientation, which it did not.
- Line 256: Why do the methods state "where the pAH is a crystallization control agent⁴¹ essential for infiltration (Supporting Information 13)." We used polyacrylic acid for the CaCO₃ mineralized collagen, so pAH is not essential (M. J. Olszta, E. P. Douglas, L. B. Gower, Scanning electron microscopic analysis of the mineralization of type I collagen via a polymer-induced liquid-precursor

(PILP) process. *Calcif. Tissue Int.* 72, 583-591 (2003).

- CaCO₃ system: I found it interesting that you guys apparently never got aragonite either; that was what we really wanted since we figured it would lead to the nice anisotropic growth like HAp. It was interesting to see the calcite disks were a bunch of little nanoparticles. Makes sense since there isn't exactly a slot across the fibril to have a full disc, even though ours had the perplexing appearance of orthogonal disks which apparently just wrapped all around the organic matter in between.

- pg. 15, line 336: "We then demonstrate that the gap regions - where the HAp crystals nucleate 12 - contain discrete, elongated channels, and that the first crystals to form have similar dimensions and are needle-shaped." I don't think you showed convincing evidence of needle-shaped crystals. Perhaps the bone was too mature; but you shouldn't be pushing to try and fit with every hypothesis out there. As I said above, your movie S5 shows better evidence of that.

- pg 15: Discussion on the confinement model. You seem to be leaving off an important aspect, and simply discussing this as crystal growth, when you know in your prior Nudelman paper, and in the bone papers by Mahamid, that the crystals are growing from within an amorphous calcium phosphate precursor. Given that ACP is already seen throughout the fibril before the streaks of crystals are, this argument of crystals growing and pushing aside the collagen in the denser overlap zones seems inconsistent. It is the ACP phase that is already confined, is it not? Then if the crystals grow within the ACP, it would lead naturally lead to the organization you find. This was nicely argued in this paper, which deserves mention (Y. Li, C. Aparicio, Discerning the Subfibrillar Structure of Mineralized Collagen Fibrils: A Model for the Ultrastructure of Bone. *PLoS ONE* 8, e76782 (2013).

- pg 17, line 402: "our model does not address the higher levels of hierarchical organization of the mineral platelets^{2,3}" I think you need to be citing some of Frank Tay and Pashley's work since they have shown the interesting hierarchical crystals with a banding pattern more like bone, highlighting the potential relevance of the NCPs and/or some phosphate moieties in causing that banding which the pAsp model doesn't (or does at the early stage).

- Figure 4: It appears that the collagen fibrils have lost the banding pattern. This is something that in the CaP system would suggest the fibril has become infiltrated with an amorphous phase that has not yet crystallized. However, the SAED pattern does not seem indicative of this, but I don't know how big a region the pattern was taken from. If you think this is not the case, then what would be causing the disruption of your collagen banding, and might that not be the reason it is not becoming very highly mineralized with the iron oxide?

Laurie Gower

Professor of Materials Science & Engineering

Reviewer #3:

Remarks to the Author:

The manuscript by Sommerdijk and colleagues presents a detailed study of crystal organization in collagen fibrils using electron tomography. The authors conclude that the model of mineralized collagen in which the plate-shaped crystallites are organized in stacks of the same orientation over a long range is incorrect and that the platelets are arranged in stacks of 4 to 8 and that they are only uniaxially co-aligned. Furthermore, the authors hypothesize that the crystal growth inside the fibrils is not guided by biological molecules but limited by physical confinement. The manuscript is well written and the main conclusion is well supported by the data. There are a few issues that should be clarified.

The story of channels is quite confusing, what are 1D molecules and 1D and 2D channels? The molecules have high aspect ratio but they are 3D objects. Similarly with the channels they are 3D. But semantics aside, it is really hard to understand what the 2D described on 12 and Fig 1a; the only channels I can see there are the gaps. But these are not parallel but perpendicular to the collagen molecules' axes. The authors also suggest the model in which the mineralization starts in

gaps was never confirmed empirically. It is not correct, it is a common knowledge that mineralization starts in gaps. In their seminal 2010 paper the authors show that ACP particles accumulate in gaps and present the explanation that the mineral is "sucked in" by positive charges of a-band. The authors did not propose templating per se, as Silverman and Landis did, but definitely there are interactions between collagen and mineral.

Another question - if the c-axes grow along the path of less resistance, why then they do not grow in gaps perpendicular to the fibril axis, but rather burrow into the channels?

It is also not clear if the X-ray based model accounts for water. It does not seem so. There is quite a bit of water in and around fibrils which gets displaced during mineralization.

I would also suggest the authors to incorporate discussion of two dark field studies by Larry Arsenault on turkey tendon (*Calcif Tissue Int* (1988) 43:202-212) and recent paper by Henry Schwarcz (*Bone* 135 (2020) 115304) into their discussion.

Reviewer #1 (Remarks to the Author):

Review on "Intermolecular Channels Direct Crystal Orientation in Mineralized Collagen" by Y.F. Xu et al.

General Remark 1-1

The authors describe a mechanistic model for HAP plates generation in bone. They state that 2 nm channels in the gap region of collagen serve as template for initial HAP needle seeds oriented [001] along the channel. Further, lateral growth of the needle induces a plate formation by keeping c-axis orientation parallel to collagen fibril long axis.

Chemical templating or involvement of non-collagenous proteins is not needed for the explanation of the bone composite material formation. Further on, they use γ -FeOOH and CaCO₃ to show the general principle of their model.

The work is excellent presenting a lot of investigations including X-rays, cryo TEM on bone and on a model system comprising in vitro mineralized collagen.

Author response:

We appreciate the reviewer's positive and encouraging comments on our study.

Detailed Remark 1-1

Length of channels known?

Author response:

Yes. As we have described in the paper (Line 175-176), the channels span the gap zone but have a discontinuity between the e2 and e1 bands, which separates them into discrete sections of 15-20 nm in length. The manuscript contained a typographical mistake indicating the discontinuity to be between 'e1 and d bands'. This has now been corrected.

Detailed Remark 1-2

Is it possible to have different HAP needles along the channels, and are they single crystals?

Author response:

The channels are only 2~3 nm in diameter, but 15-20 nm in length, which would allow more than one single crystalline needle to form end-to-end in one channel. The needles are indeed most likely single crystalline, considering their shape and extremely small sizes. As it is difficult to directly confirm this using cryoTEM, we hesitate to make this statement in the paper.

Detailed Remark 1-3

Did the in vitro-mineralized collagen aspartic acid serve as non-collagenous inhibitor?

Author response:

Yes. We now mention this in Line 186-187, and for clarity we use the more general term 'crystallization control agent', and references of the original method have been added.

Line 186-187

..where pAsp was used as a crystallization control agent following the method developed by Gower et al.^{15,16}

Detailed Remark 1-4

How does the low pH of the acid influence the mineralization process compared to mineralization without? Formation of acidic CaP (Brushite, octacalcium phosphate) preferred?

Author response:

We performed the mineralization within a HEPES buffer with pH=7.4 (please see Materials and Methods: Mineralization with Hydroxyapatite), so the adding of pAsp did not affect the pH value of the reaction.

Detailed Remark 1-5

Did the authors observe other calcium phosphate phases such as tricalcium phosphate, brushite or octacalcium phosphate (for literature see e.g. Vyalikh et al. Early Stages of Biomineral Formation—A Solid-State NMR Investigation of the Mandibles of Minipigs. Magnetochemistry 2017, 3, 39; doi:10.3390/magnetochemistry 3040039)? In the electron diffraction, around the primary beam the small angle reflections are over exposed thus the observation of these phases remains difficult e.g. the 100 reflection of octacalcium phosphate. On the other hand, it is clear that needle shape crystals should be HAP, but at initial state also metastable phases could be formed (OCP e.g. in young bone and dentine).

Author response:

While we only identified the HAP phase in this study, it is indeed possible that other crystalline calcium phosphate phases form as precursors in the early stages of collagen mineralization in bone (after the formation of ACP, as will be discussed below) as indicated by previous spectroscopic detection of octacalcium phosphate (Crane, N. J., et al. Bone 2006; Vyalikh et al. Magnetochemistry, 2017), and an acidic disordered calcium phosphate (Akiva et al., JACS, 2016).

We showed in previous work that OCP also can act as a precursor in *in vitro* HAP formation (Habraken et al., Nat. Comm., 2013) but cannot be reliably identified using cryoTEM and low dose electron diffraction, especially given the small size of the needles and their sparse occurrence in the early stages in the present system. Although intriguing, resolving this would need further studies using synchrotron X-ray scattering or NMR that fall outside the scope of the current study. In order to prevent misleading, we have mentioned the possible existence of ACP and OCP in our samples in Line 124-128, and cited the reference suggested by the reviewer.

The manuscript now reads:

Line 124-128

No other calcium phosphate phase was detected, although it is possible that the sample still contains small amounts of amorphous calcium phosphate (ACP) and octacalcium phosphate (OCP), which have been suggested to be precursors to bone mineralization^{28,29} and can still exist in mature bone,^{24,30} but are difficult to identify by LDSAED.

Detailed Remark 1-6

Why do not appear the initial needle at all plates? Is it etched away or dissolved during the growth procedure or is it simple too small to be recognized in the final HAP plate? HAP plates can have also roundish edges, thus the initial needle should form a straight linear edge and thus identified probably in the plate?

Author response:

The reasons suggested by the reviewer could indeed explain why the initial needle does not appear at all plates. The size of the needles, and their contrast in low dose electron tomography is such that we only visualize them when they were protruding out of a developing platelet, not as individual needles in the collagen matrix. Unless widening of the needles occurs, by growth along the a-

axis, they will be etched away in an Ostwald ripening process. As this widening will occur through a crystal growth process there likely will be no trace of the initial needle once it has developed into a platelet. Hence there is only a small time window in the crystal growth process where these needles could be recognized with electron tomography. It is really visualizing a “needle in a hay stack”.

And indeed the ‘needles’ are not roundish edges of HAp platelets, but have ~10 nm long linear edge as we show in Supporting Information 5. It is however difficult to clearly visualize the intrafibrillar HAp platelets that are perpendicular to the (100) face due to their thickness (only ~3 nm). We have emphasized the linear edge of the needles in Figs. 6e and 6f.

Detailed Remarks 1-7

If the mineralization starts in the gap region, does it mean that the overlap region remains unmineralized?

Author response:

The HAp crystals we observed in our experiments are 60~65 nm in length, which is close to the length of the 67 nm periodical structure of collagen (overlap+gap region). Therefore, the crystals have to propagate from the gap region into the overlap region by pushing aside the collagen molecules, in accordance with the proposed mechanism.

Detailed Remarks 1-8

Did the authors tried to isolate the needles from the collagen by dissolving it at the initial state of mineralization? In this way they would have a direct prove for their model in Fig. 6b and show in HRTEM individual HAP needles as prove!

Author response:

This is a good suggestion, but in practice, very difficult to accomplish. As indicated above, it is very demanding using cryoTEM to find (and distinguish) the small quantities of 2x15 nm rods formed at the early stages, after dissolving the collagen. Furthermore, using HRTEM would involve the drying of the sample which will create other artefacts, including the precipitation of calcium and phosphate on the needles, potentially altering their shape and structure.

A long term strategy that we are now developing is the use of liquid phase electron microscopy, with the aim of live observation of the nucleation process in a thin collagen fibril. This however is an undertaking with many technological hurdles, which we do not expect to overcome in the coming year. It therefore falls outside the scope of this manuscript.

Detailed Remarks 1-9

Besides HAP and others also the possibility of the formation of amorphous calcium phosphate has to be considered as discussed e.g. in an actual paper by Swarcz et al. in Bone 153, (2020) 115 304. They also discuss the topic which is widely held in theory that the gap zone should reside the most of the mineral in bone. Can the authors exclude formation of amorphous calcium phosphate in the channels at least in the early stage of mineralization? Of course, it is difficult to prove by electron diffraction since it gives no signal (reflections) in the ED patterns. Thus, applying of HAADF or EELS/EDS could only help in TEM. The presence of (suprafacial) amorphous calcium phosphate on HAP platelets was

shown with NMR by von Euw (Bone mineral: new insights into its chemical composition. Scientific Reports (2019) DOI: 10.1038/s41598-019-44620-6)

Author response:

We fully agree with the reviewer that ACP should form at the early stages as a precursor, and that was indeed what we observed in an earlier study (Nudelman et al., Nat. Mater., 2010). The reviewer is quite correct that we did not emphasize this precursor and its transition into HAp in our manuscript. We have now added discussion about ACP to our text in Line 124-128 (please see our response to Detailed Remark 1-5), Line 360-364, Line 373-390 and Line 410-417. The 2020 paper of Schwarcz et al. has been discussed in Line 410-417, while the 2019 paper of von Euw et al. was cited in Line 127 as well as our discussions about the amorphous layer on HAp platelets in Line 339.

Interestingly, the 2020 paper of Schwarcz et al. claims that there is mainly ACP in the gap region of mineralized collagen, which is obviously different from our observations, or other studies on this topic. The origin of this discrepancy may be related to the fact that only a small fraction of the diffraction signal was used to form the DFTEM images in that study. As HAp crystals are uniaxially and imperfectly oriented within the mineralized collagen fibrils, and can be twisted as shown in our study, most of the crystals may have no contrast in the DFTEM images. Therefore, a significant fraction of the material identified as ACP in that paper might not be truly amorphous, but instead may not diffract at the angle of observation. This has been discussed in our text (Line 410-417)

We have included the reference and the following discussion in the manuscript text which now reads:

Line 360-364

Based on our previous observations of an *in vitro* model system, which show that mineral infiltration in the form of ACP begins in these sites,³⁵ we propose that the first crystals form by the transformation of the ACP in the channels, resulting in the formation of needle-like crystals whose sizes and shapes are defined by the geometry of the channels.

Line 373-390

Additionally, it has been shown both *in vivo*²⁸ and *in vitro*³⁵ that collagen mineralization proceeds through an amorphous calcium phosphate precursor that subsequently transforms into crystalline HAp. In line with these observations, we propose that ACP readily fills the channels in the gap regions (Figure 6b), and that smaller amounts of ACP will infiltrate into the tightly packed overlap regions where intermolecular spacings are <1 nm (Figure 6a). This is likely to be a synergistic process where the collagen defines the shape of the ACP, and the ACP infiltration induces some distortion of the collagen structure, as also observed for CaCO₃ (Figure 5g-i). A population of randomly oriented HAp nuclei will then form in the channels, either *via* dissolution-recrystallization, or pseudomorphic transformation of the ACP precursor (Figures 6c).⁵³

Based on previous *in vitro* experiments using cylindrical nanopores²⁵ we expect only those crystals oriented with their fast growing *c*-axes aligned with the channel (and thus the fibril) axis are able to grow unrestricted into needle-shaped crystals (Figure 6d). The remaining ACP and the smaller crystals that are oriented in other directions will subsequently be consumed by the growth of the needles, either by a dissolution-reprecipitation process (Ostwald ripening),⁵⁴ or by adjusting the ionic arrangement and fusing with the growing HAp crystal.⁵⁵ The lateral growth of the needles will then generate platelets by pushing aside neighbouring collagen molecules (Figure 6e).

Line 410-417

We emphasize that the final intrafibrillar mineral principally comprises crystalline HAp nanoplatelets, rather than needle-like crystals⁵⁶ or ACP⁵⁷ as suggested by some dark-field TEM (DFTEM) studies. These discrepancies may be related to the fact that only a small fraction of the total diffraction signals were used to form DFTEM images. As the intrafibrillar HAp nanoplatelets have an imperfect uniaxial orientation and can also be twisted, most will have no contrast or only be partly visible in the DFTEM images, making it difficult to deduce their crystallinity and morphology with this technique.

Reviewer #2 (Laurie Gower, Professor of Materials Science & Engineering)

We would like to thank Laurie for putting such a huge amount of time into reviewing our manuscript. She has done a fantastic job, and her comments have genuinely helped us to make some very important improvements to our manuscript.

General Remark 2-1

This paper is interesting and well written with only a few minor errors (see detailed comments), and I agree with the main conclusions that this data suggests that no specific interactions with biological macromolecules are required to give similar crystal faces and orientations as bone, and agree that much of the structure can be reproduced by general physical chemistry. Of course I agree because I have been making that same argument for years. This was already demonstrated in the Olszta paper several years ago that developed the first model system. Then the Cantaart/Meldrum paper showed quite definitively that hydroxyapatite grown under confinement will inevitably lead to the same [001] orientation as bone. So to me, no revolutionary new concepts have come from this work. Nevertheless, this paper elegantly demonstrates the concepts further by examining two other, non-bone mineral systems. I like this idea, and in fact had come up with the same idea several years ago and submitted an NSF proposal to do exactly the same thing, but I had no success in getting it funded. So I'm glad the authors did it here, and especially since they are so much better equipped with the microscopy instruments and skills. Although I think Meldrum already hit the nail in the coffin with the Cantaart paper by demonstrating that HAp will obligingly form oriented crystals with sufficient confinement, it seems other groups keep arguing for protein epitaxy relationships no matter what. So maybe this paper with its lovely images and dual systems will help overcome that unrelenting belief, so that no more grant money has to be wasted. So for this reason, even though I didn't learn that much in this paper, I think others might. Having said that, I have several suggestions which I think will make this paper more impactful.

Author response:

Our work is indeed based on the pioneering work of her and her colleagues (Olszta et al., 2007), in which they successfully mineralized collagen fibrils *in vitro*, and suggested that the HAp platelets are uniaxially oriented. We hope our study will contribute to close the long standing debate that Prof. Gower and her colleagues started on the origin of HAp crystal orientation in collagen.

General Remark 2-2

This paper nicely models the channels within a collagen fibril, but I wouldn't say they have proven that, as the title of the paper implies. In contrast, the recent Reznikov and Kroger paper seems to demonstrate that the crystals meander in

and out of the neighbouring collagen fibrils, so there seems to be a disconnect between your hypothesis and their observations of bone. Just showing a model of extra space in the gap zone doesn't actually prove that the crystals are forming within any predefined channels, especially considering the crystals grow beyond those supposed channels into the overlap zones

Author response:

We emphasize that the gap zone channels are not visualizations of computer modelling, but from X-ray diffraction data of the unmineralized, hydrated type-I collagen fibril. We do therefore consider this as proof of the existence of the channels in the gap zone. As **1**. The channels are the dominating space available within the gap zone and **2**. Nucleation has been demonstrated to start in the gap zone, by association we conclude that HAP nucleates in the channels.

We fully agree with the phenomenon observed by Reznikov et al. that HAP crystals can exit/enter collagen fibrils, which was in fact also observed in our experiments (Supporting Information 3). We however do not think this conflicts with the model that we propose. The c-axes of the HAp crystals are not perfectly aligned along the collagen fibril axis, which means the misoriented crystals can propagate into other fibrils when they grow longer, even if they nucleated within the channels in the gap region of collagen fibrils.

General Remark 2-3

Having said that, I think you might actually data that does show some evidence of channels in your video movie S5. In this movie of the isolated model system fibril, there are some interesting patterns if one slowly moves the frames back and forth as you focus on the dots at the outer edge of the fibril. I can see "streams" of black dots that appear to extrafibrillar, and then they often take a sharp curve inward as they infiltrate into the fibrils. I suspect these dotted streaks are still actually amorphous nanoparticles (or PILP-like nanodroplets) that seem to line up prior to entry into the fibril (as was previously shown in your Nat. Mat. 2010 paper). But in this movie one can see that the connected dots seem continue along a connected path that infiltrates into the fibril, appearing to be a channel. This data might even support the capillary infiltration mechanism.

Author response:

The reviewer refers to the cryo-tomogram of the *in vitro* mineralized collagen fibril. This fibril has a diameter of ~300 nm and thereby has significantly lower contrast than the bone sample which has only a thickness of ~100 nm, or the collagen infiltrated with lepidocrocite, which was measured in dry state. As a consequence the dotted appearance of the mineral may very possibly be due to the low signal-to-noise ratio in the images. This interpretation is supported by the observation that at this state the HAp crystals have clearly developed into platelets already, as can be seen in Movie S6.

We therefore prefer a careful and conservative, rather than a speculative interpretation of the data.

General Remark 2-4

However, I understand if you don't want to go into infiltration mechanisms here since this paper is focused on orientational growth inside the fibril. But it does nicely illustrate the interesting channels of entry, which could perhaps be commented on without trying to pronounce any particular mechanism, yet supporting the argument that there are some type of channels. Whether those channels are pre-existing or not may remain to be determined. But I assume you could determine where these channels are starting along the fibril gap zone or specific band as you did in your prior paper, and a consistent starting location

might suggest a consistent entry channel position. This might also fill the gap between the Reznikov observations and your model system, because it shows a pre-crystal that spans the outside and inside of a fibril, all connected by one dotted path.

Author response:

We have now labelled all of the bands in Fig.3c. These indeed show that the channels start in the a-band region, in line with our earlier report.

We adjusted the manuscript now to read:

Line 359-360

We then demonstrate that the gap regions - where the HAp crystals nucleate¹² - contain discrete, elongated channels that **start in the a-band region.**

General Remark 2-5

I think your section starting at line 344 on the confinement-based model is inconsistent with your prior work, as well as bone studies, because you seem to be describing crystal growth as entering into the overlap zone, when your prior work, as well others, shows there is amorphous precursor throughout those regions as well, prior to the streaks of crystals. So the confinement is really of the ACP precursor phase, and not crystals pushing there way into the denser collagen region, right? Crystals growing from within an amorphous phase is also a form of confinement, so if the ACP is confined first, then the crystals will likely be as well. Especially if it follows roughly a pseudomorphic transformation (although dissolution and recrystallization can't be ruled out). In any case, I think you need to rework that section in remembering there is an ACP phase present that is initially confined.

Author response:

We agree that ACP plays an important role during the mineralization of collagen. The discussions about the role of ACP have been added to the text in Line 125-129 (please see our response to Detailed Remark 1-5) and Line 360-364, 373-390 and 410-417 (please see our response to Detailed Remark 1-9). And the formation of ACP at early stage has been illustrated in Figure 6b. Please also see details in our response to Detailed Remark 2-20.

However, in our opinion, it is unlikely that the final morphology of the HAp platelets is identical to the shape adopted by the infiltrated ACP phase. HAp has a very strong preference to form as either needles or platelets. Even within collagen these are the morphologies that are observed (or variants of these).

This is supported by our SAXS/WAXS data on CaCO₃ (Figure 5g-i), which show that although the infiltration of ACC phase has slightly distorted the collagen structure, a more significant distortion happened upon amorphous to crystalline transition. Based on these data and the structure of collagen gap/overlap regions, we propose that the HAp crystal platelets are not directly transformed from initially infiltrated 'ACP platelets', and the growth of the HAp crystals have further distorted the collagen structure, as shown in our model in Figs. 6e and 6f.

We have added a more detailed description of our SAXS/WAXS data in Line 274-291, and modified the text in Line 377-380 and Line 395-397.

Line 274-291 :

No changes in the SAXS and WAXS spectra were observed during the first 80 minutes of the reaction, while an increase in the scattering intensity was detected after 90 minutes (Figure 5g). This can be attributed to the formation of

amorphous calcium carbonate (ACC) *within* the fibrils, where this causes a small decrease of the lateral packing distance of the collagen molecules. This is demonstrated by broadening of the SAXS peak at $q = 4.08 \text{ nm}^{-1}$ ($\sim 1.5 \text{ nm}$), which corresponds to a reduction in the separation of the collagen molecules (Figure 5h and 5i).

WAXS confirmed the development of both vaterite and calcite at incubation times $> 95 \text{ min}$ (Figure 5g), while SAXS revealed that crystallization of the ACC is accompanied by a significant reduction in the intermolecular distances of the collagen molecules from ~ 1.5 to 1.1 nm , as shown by the replacement of the $q = 4.08 \text{ nm}^{-1}$ peak with one at $q = 5.68 \text{ nm}^{-1}$ (Figures 5h and 5i). The axial $\sim 67 \text{ nm}$ d-band organization remains unchanged during this process. The amount of crystalline material in the sample then continued to develop with time and significant amounts of calcite and vaterite were observed after 440 minutes (Figure 5g). Similar observations have been made for bovine⁴⁷, fish bone,⁴⁸ and turkey tendon mineralized with HAp,⁴⁹ demonstrating that the collagen molecules are pushed apart and compressed by initial infiltration of the ACC, and more significantly by its subsequent crystallization.

Line 377-380

This is likely to be a synergistic process where the collagen defines the shape of the ACP, and the ACP infiltration induces some distortion of the collagen structure, as also observed for CaCO_3 (Figure 5g-i).

Line 395-397

This is supported by our SAXS/WAXS data on CaCO_3 (Figure 5g-i) that show the rearrangement of the collagen during the amorphous to crystalline transition.

General Remark 2-6

The final schematic (Fig. 6) is nice for representing the random lateral orientation of the crystals, but one can see that in the actual images anyway. The use of a schematic would be better for illustrating the orientation mechanism, which is the crux of your story, and one which the community doesn't seem to have grasped yet. It should show how the crystal nuclei presumably start off more random, but then become oriented as the rapid growth direction takes over. In addition, you show the same rectangular platelets as the old literature, but a key part of your message is that they twist as they traverse down the fibril; so I think your schematic should show that as well. And relating to my prior comment, I think the interstices should show an amorphous phase from which the crystals nucleate and grow, rather than crystal tips growing into the space.

It seems you are focused on the argument that it is these crystal tips that have fooled people into thinking the crystals are needles, but this part of your story is very weak. I'm pretty sure this is not the cause of the needle-like observations by many others, and certainly not in the recent report by Reznikov who tracked the crystal trajectories and did not make the mistake of thinking the tips of a growing platelet was a needle. They argue that there is fusion of the originally needle-like crystals. Once again, you do have data that could better illustrate that, because the movie S5 shows a more needle-like morphology. This is presumably early stage mineralization since the crystals streaks are quite dotted, and apparently the nanoparticles/droplets are not fully fused and/or crystalline. So in my opinion, I think this section could be reworked, and you could do a better job in this

schematic of illustrating how the crystals become uniaxially oriented but twist, as well as illustrating what is actually confined (the ACP phase), and how the ACP phase and early stage crystals infiltrate and are more needle-like.

Author response:

We have revised Figure 6 accordingly to show these details including the twisting of the crystal platelets and the formation of ACP at the early stage, and discussed them in the text in Line 373-390 (please see our response to Detailed Remark 1-9) and Line 392-402.

We have addressed the concerns about Movie S5 above (please see our response to General Remark 2-3).

Concerning the needle-shaped tip of the HAp platelets, we want to emphasize that we are not trying to correlate our observations with all the older papers that observed 'needles' (e.g, the 1988 paper of Arsenault et al.), as they very likely observed edge-on view platelets as suggested by the reviewer, and/or misinterpreted the DFTEM results. We have made this more clear in the text (Line 410-417, please see our response to Detailed Remark 1-9).

This is exactly why we cited the papers of Reznikov et al. for this discussion, as our observations and interpretation are quite similar to those of Reznikov et al. (needle shaped tip of HAp platelets). The only other paper that we cited there is the 1992 paper of Traub et al, in which it was emphasized that stereo pairs of TEM images showed that in the early stages of collagen mineralization the crystals had needle shapes, different from the edge-on views of crystal platelets observed by others. More importantly, the paper proposed for the first time that the needles would later develop into crystal platelets. We have described the findings of these two papers more precisely in (Line 404-410).

The manuscript now reads:

Line 392-402:

This mechanism is possible due to the unique structure of collagen, where the gap region is more compliant and compressible than the overlap region.⁵⁶ Continued growth of the crystals along the *c*-axis will cause them to extend into the overlap region, causing a rearrangement of the collagen molecules, and a concomitant translocation of the included ACP. This is supported by our SAXS/WAXS data on CaCO₃ (Figure 5g-i) that show the rearrangement of the collagen during the amorphous to crystalline transition. During this process, the platelets also start to twist along their long axes in agreement with previous reports (Figure 6f),² which we tentatively attribute to the locally changing collagen organization within the microfibril. This eventually leads to twisted HAp platelets that are uniaxially oriented along their *c*-axes, where adjacent platelets may direct each other and form small stacks, depending on the degree of mineralization (Figure 6g, see also Figures 2f and 2n).

Line 404-410:

This scenario is in excellent agreement with studies of fish bones¹⁹ and human bones,² which showed that HAp nanoplatelets^{2,12} only form short or irregular stacks, that may become intergrown in time.² Furthermore, the recent study of Reznikov et al. also observed needle-like tips on HAp platelets in human bone that resemble 'fingers of a hand' as well as individual needle-like crystals,² and also proposed that the HAp nanoplatelets develop from these needle-like crystals. We note that a similar scenario was suggested as far back as 1992 by Traub et al., based on the observation of needle-like crystals in the early stages of turkey tendon mineralization.¹²

Detailed Remark 2-1

I love the videos of the fibril cross-sections which nicely illustrate the propeller like orientational rotation of the crystals along the length. The explanation and demonstration of why the old Landis reports only saw biaxial oriented crystals was nice, and in that case was respectful.

Author response:

We appreciate the reviewer's positive comment.

Detailed Remark 2-2

Pg 3, line 75. "it has been suggested that the orientation of the crystals may be governed by their confinement within the gap zone of the collagen.^{15,18} However, for such a mechanism to operate the gap zones could not contain the flat, 2D channels present within the traditional model of collagen, as these would in fact allow HAp to grow with its c-axis in any direction parallel to the channels, rather than preferentially along the long axis of the fibrils.¹⁷" The Olszta paper did not claim the crystals were confined only within the gap zone. Quite the opposite. Others suggested the crystals may nucleate in the gap zone, but our argument was that the precursor might infiltrate there, but the crystals clearly could outgrow the gap zone and we proposed the rapid growth direction of HAp leads to orientation. Which is exactly what you are arguing for here. I think you are just trying too hard to prove you have come up with something new, but its mostly not, just much more nicely illustrated.

Author response:

We apologize for the inaccurate description of the work of Olstza et al. Indeed, the work of Olstza did not claim that the crystals were confined within the gap zone. In contrast, it proposed that ACP infiltrated through the collagen fibril (also in the overlap region), and a rapid growth of HAp then induced the uniaxial orientation of the HAp crystals. The main difference between the Olstza model and the one we propose is that the former is based on an simplified collagen model, in which straight collagen molecules are (visualized as) rods packed in quasi-hexagonal arrays, and the gap region was plotted as 'empty volumes'. Such a structure is not consistent with the X-ray scattering results of the collagen structure, and would allow HAp crystals to grow in any direction between the collagen molecules arrays rather than uniaxially oriented (as pointed out in the PNAS paper of Tao et al., 2015). Our current model has updated this pioneering model, and we hope it will make the confinement-based mechanism better accepted by the research community.

We have now described the contributions of Olstza et al. more precisely in Line 73-76 and Line 319-326, and emphasized the difference between our collagen model and the traditional quasi-hexagonal model in Line 328-331.

The manuscript now reads:

Line 73-76

As an alternative, Gower and coworkers suggested that the orientation of the crystals may be governed by their confinement within the collagen fibrils,¹⁵ which was supported by the finding that in vitro cylindrical nanopores can direct the oriented growth of the HAp crystals.²⁵

Line 319-326

Reznikov et al. did not discuss the origin of crystal orientation with respect to the collagen fibrils while Burger et al. proposed that parallel 2D channels exist within a collagen fibril (cf. Figure 1a), which could template the formation of HAp

platelet stacks. Deformation of the matrix during this process would also lead to distortion of the parallel alignment of the crystals.¹⁹ Olstza et al. proposed that the HAp platelets form by transformation of the amorphous calcium phosphate confined in the space between collagen molecules, but did not yet have the details of how this intermolecular space was distributed throughout the collagen structure.¹⁵

Line 328-331

While the community had considered the collagen structure to consist of a quasi-hexagonally packed array of staggered, straight collagen molecules, in 2006 the X-ray diffraction study of Orgel et al. revealed a more complex triclinic superstructure, in which the tropocollagen molecules are organized in a twisted and tilted arrangement.³¹

We also have expanded and refined our illustration in Figure 6 to demonstrate the different steps in which the gap zone channels direct the mineral growth.

Detailed Remark 2-3

pg 4, line 93: "This provides compelling evidence that the orientation of HAp within collagen fibrils arises due to confinement effects only." I think the compelling evidence was already provided by the Cantaart model system, which shows the HAp will show preferred orientation in constrained nanopores of non-specific interactions, period! That should have been the end of this debate.

Author response:

While the paper of Cantaert et al. indicated that confined growth of HAp in the cylindrical nanopores of track-etched membranes can induce a preferred orientation, it did not demonstrate that such a mechanism also applies for the mineralization of collagen fibril, as the latter is a pliable matrix with much more complex structure. This motivated us to perform this detailed study, and we hope our findings can indeed help to end this debate. We have clarified this in the text:

Line 76-77:

However, it was not clear if a pliable matrix of collagen fibrils could similarly constrain crystal growth

Detailed Remark 2-4

Fig. S2. I find your wording on the captions to be a little confusing. I would have thought a "top view" of the fibril would be like a top down view (looking down the c-axis). Seems like these are side views of the longitudinal axis of the fibril. I guess you are referring to top view of stacks, but some clarification might be useful since not everyone is well versed with tomographic techniques.

Author response:

We have now defined in the legend that tomography reconstruction slices viewed perpendicular to the collagen fibril long axis as 'z-slices', while tomography reconstruction slices viewed along the collagen fibril long axis as 'y-slices', and changed the text/figure captions accordingly.

The following text has been added:

Line 103-105

Computer generated lateral tomography reconstruction slices viewed perpendicular to the long axes of the collagen fibrils are defined as 'y-slices',

while the longitudinal reconstructed slices viewed along the fibril long axes are defined as 'z-slices'.

And the legend now reads:

Line 797-799

Tomography reconstruction slices viewed perpendicular to the long axes of the collagen fibrils are defined as 'y-slices', while the slices viewed along the fibril long axes are defined as 'z-slices'.

Detailed Remark 2-5

pg. 6, line 129: "and usually propagated into the fibrils (Supporting Information 3)." This data is not convincing unless one knows for sure those are two different fibrils; I'm not sure how that was known other than the color overlay you provided. Not that I doubt that occurs, given the platelet tracking done in the Reznikov paper. But this statement might raise some brows.

Author response:

We apologize to the reviewer that we did not make it clear that the two different collagen fibrils shown in Supporting Information 3 are the same two fibrils that we show in Fig. 2a and 2e. By viewing the tomography in two directions, we demonstrate that these are two different collagen fibrils. We have now clarified this in Supporting Information 3.

Supporting Information 3 now reads:

Figure S3 tracks one HAP platelet through different cross section slices of two adjacent collagen fibrils (as shown in Figs. 2a and 2e).

Detailed Remark 2-6

Fig. 3: It might be nice to put markers of where the e1 and d bands on the schematic since they are referred to in the text. (and maybe channel entry points as mentioned above).

Author response:

We thank the reviewer for the suggestion and have labelled all the bands in figure 3. As mentioned in our answer to Detailed Remark 1-1, the discontinuity is between e2 and e1 bands. This typo has been corrected in Line 175-176.

Detailed Remark 2-7

pg. 8, line 171: "and are markedly different from the traditional picture which depicts 2D channels that run parallel to the fibril axis (Figure 1).1,13,18,30" I don't think anyone ever thought that only a rectangular channel is formed in 2D. We all know there must be longitudinal channels, but it is just difficult to draw in a schematic. Now we know they are slightly tilted, so even harder to draw. Even your collagen in Figure 3 shows something that looks like a 2D rectangular region of channels. Likewise, schematic in Fig. 6 doesn't show longitudinal channels either beyond the gap zone, so I don't see why you are saying your perspective is markedly different.

Author response:

We apologize for the confusion. The structure of unmineralized collagen was depicted as staggered layers of collagen molecules in the early literature (e.g., Landis et al., J. Struct. Biol., 1993). These would clearly leave rectangular channels in the gap region, as is also shown in our Figure 1. This representation

of 2D channels was used also in the 2008 paper of Burger et al., as well as some recent papers (e.g., Kim et al., Nat. Comm., 2018, <https://www.nature.com/articles/s41467-018-03041-1>). A different model was presented in the paper of Olstza et al., which proposed HAp growth in both gap and overlap regions, as discussed above. In our current model, we show that there are 2–3 nm wide, 15–20 nm long cylindrical channels within the gap region of collagen. There are indeed some small lateral connections between the channels, which are however only ~0.3 nm in width and therefore will still constrain the crystal growth in lateral direction. In contrast, all the open spaces in the overlap regions are <1 nm and would inhibit the nucleation of HAp crystals. Our model is therefore markedly different from both the Landis' and Olstza's model. We have now clarified these differences and described the small connections between the channels in Line 176-178, and highlighted them in Figure 3 in order to prevent confusion.

The manuscript now reads:

Line 176-178

In lateral directions, these channels are mostly separated from each other, with only ~0.3 nm wide small connections (Figure 3b, magenta arrow).

Detailed Remark 2-8

pg 6, line 146. I think comparing the tip of a platelet as being related to literature describing needles is not something to be correlated. Those reports were usually looking down the length, and not at the tip of a platelet. It is more likely the old edge-on view, and/or early stage mineral.

Author response:

We respectfully disagree with the referee here. As discussed in our response to General Remark 2-6, our observation of needle shaped tips on HAp platelets is quite similar to those of Reznikov et al., as we now show also in Figs. 6e and 6f. So we think a direct comparison is valid.

In the case of the 1992 paper of Traub et al., the observation was indeed different: they observed needles rather than needle-shaped tips. However their conclusion was the same: they were the first to propose that HAp platelets in collagen develop from needles. We have revised our text (Line 408-410, please see our response to General Remark 2-6) to describe this more precisely.

Detailed Remark 2-9

pg. 7, line 152. "both of which limit the detection of HAp platelets that are not oriented parallel to the electron beam." I think you mean the platelets oriented edge-on were needed for contrast?

Author response:

The referee is right. We have now described this more clearly in Line 155-159.

The manuscript now reads:

Line 155-159

We note, however, that this contrast may derive from their use of thicker samples (250 and 500 nm, respectively) and of a less sensitive camera system, both of which limit the detection of HAp platelets that are not oriented parallel to the electron beam, as platelets oriented edge-on will have significantly higher contrast in TEM imaging.

Detailed Remark 2-10

pg. 8, you describe various dimensions and orientations in the in vitro model system as being similar to those in bone, but the images and videos have such a different appearance. Such as why do the regions next to the crystallites in bone appear brighter, almost a shadowing appearance, which isn't seen at all in the model system?

Author response:

The brighter regions near the crystallites in bone reflect the fact that there is an inhomogeneous density in the bone collagen fibril. Such a feature was observed in most of the TEM studies of bone sample (e.g., Reznikov et al., Science 2018; McNally et al., PLOS one 2012). The detailed reason for this inhomogeneity remains unknown.

It could be related to the sample preparation procedure (drying, embedding in epoxy, and cutting), or due to some other factors (e.g., presence of non-collagenous proteins in bone), which could both explain why this is not observed in the model systems. A direct cryoTEM study of bone would help to resolve this question, but this would be a great time investment, and outside the scope of the current paper.

We find that this inhomogeneity does not significantly affect our conclusions, as the c-orientation of the HAp platelets and the ~67 nm periodical structure of collagen fibrils are both well preserved in our samples. Furthermore, the early study of Landis et al. (which proposed the decks-of-cards model) used a similar sample preparation method.

Detailed Remark 2-11

Also, why are the in vitro crystallites seen as dark speckled platelets rather than solid streaks as in bone? Is it because it is not a slice in a dense sample; the staining that was used differs; the imaging method of stacked slices? Are the speckles because they are still amorphous nanoparticles in the early stage? I'm not doubting the data, just wish I understood the differences in imaging techniques and/or samples.

Author response:

This has now been explained in the discussions about Movie S5 (please see our response to General Remark 2-3).

Detailed Remark 2-12

Fig. 4 caption: "were measured that were at least 10 nm apart for (g) and 50 nm apart for (g)." There is no (g) or (h) images in figure 4.

Author response:

Corrected.

Detailed Remark 2-13

pg. 31, line 770: typo "magnification mage showing"

Author response:

Corrected.

Detailed Remark 2-14

pg 13, line 299: "Reznikov et al. did not discuss the origin of crystal orientation with respect to the collagen fibrils, while the models presented by Olszta et al. and Burger et al. were still based on the hypothesis that collagen fibrils contain

2D channels (cf. Figure 1a)." I don't understand why you are saying this. Reznikov tracked the collagen and crystal orientation, and showed the crystals exited and entered fibrils, so a pre-existing channel that you are trying desperately to prove may not exist.

Author response:

The study of Reznikov et al. focused on the fractal morphology of the crystals, and did not discuss the origin of HAp crystal orientation. We do not think that their results contradict the model that we propose, as explained in our response to General Remark 2-2.

Detailed Remark 2-15

And your collagen model shows the gap zone contains short channels that stacked together look quite similar to the supposed 2D channels represented in other models.

Author response:

This has been clarified above (please see our response to Detailed Remark 2-7).

Detailed Remark 2-16

Plus, you are misrepresenting the Olszta paper, given that their schematic in Fig. 5 does NOT show a deck-of-cards arrangement of the crystals, nor does it show crystals only forming in the hole zones. It was deliberately arguing against those things in the text. While I appreciate being cited for uniaxial aspects earlier, in this sentence it gives the false impression that our paper talked about 2D channels and argued for biaxial orientation, which it did not.

Author response:

We apologize again for the inaccurate description of the work in Olszta's paper, we never meant to say that a 'decks-of-cards' organization was proposed in that paper. The main difference between that paper and our current study has been discussed above (Detailed Remark 2-2), and is now also discussed more properly in the text.

Detailed Remark 2-17

Line 256: Why do the methods state "where the pAH is a crystallization control agent essential for infiltration (Supporting Information 13)." We used polyacrylic acid for the CaCO₃ mineralized collagen, so pAH is not essential (M. J. Olszta, E. P. Douglas, L. B. Gower, Scanning electron microscopic analysis of the mineralization of type I collagen via a polymer-induced liquid-precursor (PILP) process. *Calcif. Tissue Int.* 72, 583-591 (2003).

Author response:

Indeed pAH is not 'essential' for the CaCO₃ mineralization, but plays a role similar to pAA here. We have replaced 'essential for infiltration' into 'guiding mineral infiltration' in the text (Line 255).

Detailed Remark 2-18

CaCO₃ system: I found it interesting that you guys apparently never got aragonite either; that was what we really wanted since we figured it would lead to the nice anisotropic growth like HAp. It was interesting to see the calcite disks were a bunch of little nanoparticles. Makes sense since there isn't exactly a slot across the fibril to have a full disc, even though ours had the perplexing

appearance of orthogonal disks which apparently just wrapped all around the organic matter in between.

Author response:

Indeed it would make an interesting study to investigate aragonite formation within collagen fibrils, e.g. using Mg^{2+} .

Detailed Remark 2-19

pg. 15, line 336: "We then demonstrate that the gap regions - where the HAp crystals nucleate 12 - contain discrete, elongated channels, and that the first crystals to form have similar dimensions and are needle-shaped." I don't think you showed convincing evidence of needle-shaped crystals. Perhaps the bone was too mature; but you shouldn't be pushing to try and fit with every hypothesis out there. As I said above, your movie S5 shows better evidence of that.

Author response:

Following the discussions on Movie S5 (please see our response to General Remark 2-3), we have revised the description (Line 359-364):

We then demonstrate that the gap regions - where the HAp crystals nucleate¹² - contain discrete, elongated channels that start in the a-band region. Based on our previous observations of an in vitro model system, which show that mineral infiltration in the form of ACP begins in these sites,³⁵ we propose that the first crystals form by the transformation of the ACP in the channels, resulting in the formation of needle-like crystals whose sizes and shapes are defined by the geometry of the channels.

Detailed Remark 2-20

pg 15: Discussion on the confinement model. You seem to be leaving off an important aspect, and simply discussing this as crystal growth, when you know in your prior Nudelman paper, and in the bone papers by Mahamid, that the crystals are growing from within an amorphous calcium phosphate precursor. Given that ACP is already seen throughout the fibril before the streaks of crystals are, this argument of crystals growing and pushing aside the collagen in the denser overlap zones seems inconsistent. It is the ACP phase that is already confined, is it not? Then if the crystals grow within the ACP, it would lead naturally lead to the organization you find. This was nicely argued in this paper, which deserves mention (Y. Li, C. Aparicio, Discerning the Subfibrillar Structure of Mineralized Collagen Fibrils: A Model for the Ultrastructure of Bone. PLoS ONE 8, e76782 (2013)).

Author response:

We fully agree with the reviewer that the ACP phase should be incorporated into our discussions. However, as we have stated in the response to General Remark 2-5, the HAp crystal morphology are unlikely defined by the ACP phase. Considering the fact that the interstices in the overlap regions are all smaller than 1 nm, it is most likely that ACP has filled the channels in the gap regions first, and transformed into HAp nuclei (by dissolution-recrystallization, or via a pseudomorphic transformation as suggested by the reviewer), of which the growth is constrained by the channels. We have added these discussions in Line 360-364 and Line 373-390 (please see our response to Detailed Remark 1-9) and cited the references mentioned by the reviewer, and revised the scheme in Fig. 6 to indicate the infiltration of ACP phase at the early stage.

Detailed Remark 2-21

pg 17, line 402: "our model does not address the higher levels of hierarchical organization of the mineral platelets^{2,3}" I think you need to be citing some of Frank Tay and Pashley's work since they have shown the interesting hierarchical crystals with a banding pattern more like bone, highlighting the potential relevance of the NCPs and/or some phosphate moieties in causing that banding which the pAsp model doesn't (or does at the early stage).

Author response:

We have now cited these reports in the text (Line 457)

Detailed Remark 2-22

Figure 4: It appears that the collagen fibrils have lost the banding pattern. This is something that in the CaP system would suggest the fibril has become infiltrated with an amorphous phase that has not yet crystallized. However, the SAED pattern does not seem indicative of this, but I don't know how big a region the pattern was taken from. If you think this is not the case, then what would be causing the disruption of your collagen banding, and might that not be the reason it is not becoming very highly mineralized with the iron oxide?

Author response:

We are not sure about the suggestions of the reviewer here. Our in-vitro HAp mineralized collagen fibrils still have the banding pattern, although it is less visible most likely due to the heavy infiltration of HAp crystals. The banding pattern also remains visible in the FeOOH (Fig. 4a) and CaCO₃ mineralized collagen fibrils (Figs. 5b and 5d), but shows much less contrast in TEM as those samples are dried and not stained. So we think the lower mineralization degree of iron oxide is most likely related to the low solubility of iron ions in alkaline solution.

Reviewer #3 (Remarks to the Author):

General Remark 3-1

The manuscript by Sommerdijk and colleagues presents a detailed study of crystal organization in collagen fibrils using electron tomography. The authors conclude that the model of mineralized collagen in which the plate-shaped crystallites are organized in stacks of the same orientation over a long range is incorrect and that the platelets are arranged in stacks of 4 to 8 and that they are only uniaxially co-aligned. Furthermore, the authors hypothesize that the crystal growth inside the fibrils is not guided by biological molecules but limited by physical confinement. The manuscript is well written and the main conclusion is well supported by the data. There are a few issues that should be clarified.

Author response:

We appreciate the reviewer's positive and encouraging comments.

Detailed Remark 3-1

The story of channels is quite confusing, what are 1D molecules and 1D and 2D channels? The molecules have high aspect ratio but they are 3D objects. Similarly with the channels they are 3D. But semantics aside, it is really hard to understand what the 2D described on 12 and Fig 1a; the only channels I can see there are the gaps. But these are not parallel but perpendicular to the collagen molecules' axes.

Author response:

Indeed, all objects in the real world should be 3D (including the so called '2D materials'), and we are only using the terms '2D' to simplify our discussions. In this study, 2D channels refer to the rectangular channels depicted in the old collagen model as shown in Figure 1, which we have now defined as "the rectangular (2D) channels" in Line 69. And we respectfully indicate that we have not used the term '1D' to describe the channels or molecules in our paper, but only '1D-staggered' to describe the staggering of collagen molecules along the c-axis, which we think is valid.

Besides, although the 2D channels as shown in Figure 1 and in the literature examples mentioned are parallel to the collagen molecules as planes, their long axes are indeed perpendicular to the collagen molecule axes. Our previous description in Line 180 is therefore inaccurate and has been corrected to now read: "rectangular 2D channels in the gap region".

Detailed Remark 3-2

The authors also suggest the model in which the mineralization starts in gaps was never confirmed empirically. It is not correct, it is a common knowledge that mineralization starts in gaps. In their seminal 2010 paper the authors show that ACP particles accumulate in gaps and present the explanation that the mineral is "sucked in" by positive charges of a-band. The authors did not propose templating per se, as Silverman and Landis did, but definitely there are interactions between collagen and mineral.

Author response:

We respectfully conclude that there is some misunderstanding here, as we have stated in the manuscript that the mineralization starts in gap region (see Line 359 and Line 372), and discussed the possible role of collagen in inducing mineral nucleation (see Line 443-449).

Detailed Remark 3-3

Another question - if the c-axes grow along the path of less resistance, why then they do not grow in gaps perpendicular to the fibril axis, but rather burrow into the channels?

Author response:

Again, we respectfully suggest that there is some misunderstanding. As shown by our model, there is no gap perpendicular to the fibril axis within the collagen structure, and the rectangular gaps suggested by the classical model (as shown in Figure 1) do not exist, as we demonstrate. The cylindrical channels as shown by our model only have ~0.3 nm sized lateral connections (as we now highlight in Figure 3b) and therefore will constrain the crystal growth in lateral directions.

Detailed Remark 3-4

It is also not clear if the X-ray based model accounts for water. It does not seem so. There is quite a bit of water in and around fibrils which gets displaced during mineralization.

Author response:

In fact, the model was based on an X-ray diffraction study of a hydrated collagen sample in a sealed sample cell. This has now been pointed out in Line 165 of the manuscript.

Detailed Remark 3-5

I would also suggest the authors to incorporate discussion of two dark field studies by Larry Arsenault on turkey tendon (*Calcif Tissue Int* (1988) 43:202-212) and recent paper by Henry Schwarcz (*Bone* 135 (2020) 115304) into their discussion.

Author response:

In line with the suggestion of the reviewer, discussions about these two papers have been added into our text (Line 410-417, please see our response to Detailed Remark 1-9).

However, we note that these DFTEM results are quite different from other TEM/STEM studies (e.g., the current study, and Reznikov et al., *Science* 2018). For example, Arsenault et al. observed 'rod-like' crystals in highly mineralized collagen fibrils, while study of Schwarcz et al. suggested that there are mainly ACP in the gap region of mineralized collagen. As suggested in our discussions with Reviewer 1 (Detailed Remark 1-9), the origin of these discrepancies may be related to the fact that only a small fraction of the diffraction signal was used to form the DFTEM images in that study. As HAp crystals are uniaxially and imperfectly oriented within the mineralized collagen fibrils, and can be twisted as shown by our study, most of the crystals may have no contrast in the DFTEM images, or only be partly visible due to twisting, which can cause misinterpretation on their crystallinity and morphology. We have described these discrepancies and our explanation in the discussions.

Reviewers' Comments:

Reviewer #1:

Remarks to the Author:

17th July 2020

Comments of Referee 1:

Nature Communications manuscript NCOMMS-20-11225B

"Intermolecular Channels Direct Crystal Orientation in Mineralized Collagen" by Prof Sommerdijk and colleagues"

Resubmitted (rebuttal) from 9th July 2020

The authors responded point by point to my questions satisfactory, thus only two or three remarks remain:

1. Page 36, Figure 3b:

In case the scale bar is true, the triple-helices have mostly a diameter of about 1 nm showing the individual three alpha-chains, which could be resolved by X-rays. I am wondering about the discrepancy to the known value of about 1.45 nm for the triple-helix diameter from the literature. Do have the authors have an explanation for this?

2. Page 8, line 175:

The authors write in the description to Fig. 3 about a channel diameter of 2-3 nm, which gives the impression that it has a statistical variation over space. However, what they mean is that the individual pore diameter is (2x3) nm² as seen in Fig. 3b, thus indicating an elongated pore in cross section.

3. Page 39, Figure 6a:

The scheme of the principal Fig. 6a of the paper is misleading to a non-expert since it gives the impression that the macroscopic bar (collagen fibril) at top and the zoom of pores below have the same orientation with respect to each other. This would give the impression that the bar has pores oriented radial symmetric but not along the collagen fibril long axis. The authors could either indicate a cross sectional cut on the bar and a turn of 90 degrees from the bar to the zoom or they could add a bar in cross section from the beginning and indicate a zoom there.

4. Page 6 line 125-129 and page 14, line 326-336

The authors discuss on page 6 the (low) possibility occurrence of ACP and OCP in bone. They could add for OCP case a recent paper from Simon et al. in Scientific Reports 8, 13696 (2018) additional to the Vyalikh and Mahmud papers, citations 28 and 29.

Additionally, the authors discuss the chemical templating and molecular recognition of collagen and HAP citing Silver and Xu, citations 22 and 23. Here, they could also add as citation Simon et al. Scientific Reports from 2018 since the templating is directly shown in TEM by imaging individual triple-helices mineralized by HAP on the atomic level.

Reviewer #2:

Remarks to the Author:

The paper is greatly improved, and I agree with most of their revisions. However, there seems to be a major disconnect when they argue that the crystal orientation is governed by confinement in collagen channels, but then in several locations they state that the crystals push aside the collagen molecules. And they seem to argue this occurs during lateral growth, as well as crystals pushing

into the densely-packed overlap region. I think this contradictory argument could be clarified if one were to consider the confinement belonging to crystals growing within the ACP phase, and this phase is what is originally confined by the collagen during infiltration. I believe their SAXS data (which they cite as matching bone data) supports this premise.

Line 276: "This can be attributed to the formation of amorphous calcium carbonate (ACC) within the fibrils, where this causes a small decrease of the lateral packing distance of the collagen molecules. This is demonstrated by broadening of the SAXS peak at $q = 4.08 \text{ nm}^{-1}$ ($\sim 1.5 \text{ nm}$), which corresponds to a reduction in the separation of the collagen molecules (Figure 5h and 5i)."

- This broadening seems to have already taken place to a great extent before the crystallization process, where the blue curve is already broadening (as well as intensified) and the pink curve shows roughly two overlapping peaks corresponding to the initial and final state. Doesn't this suggest that the shifting around of the collagen molecules you describe takes place before the crystallization peaks emerge? (where the green and blue sharpen up as the crystals further reduce the water content and thus spacings). In other words, it seems like the collagen reorganization and thus confined space is created by infiltration of the amorphous phase. You sort of added that into your revision, except in other places you state that the crystals are pushing aside the collagen (and in your Fig. 6 schematic). For example:

Figure 6 caption: "Nuclei with their fast growing c-axes parallel to the channels out-compete those with other orientations (d), and develop into platelets by pushing aside the collagen molecules, as shown in (e)."

Line 388: "The lateral growth of the needles will then generate platelets by pushing aside neighbouring collagen molecules (Figure 6e)."

Line 392: "Continued growth of the crystals along the c-axis will cause them to extend into the overlap region, causing a rearrangement of the collagen molecules, and a concomitant translocation of the included ACP."

- What does translocation of ACP mean?

Figure 2 has a typo in the second sentence. Reconstructionlateral

Signed, Laurie Gower

Reviewer #3:

Remarks to the Author:

The authors thoroughly addressed the reviewers' comments. I recommend the manuscript for publication. I would only ask if possible for the authors to provide their opinion on the role of pAsp in in vitro collagen mineralization. Calling it a crystallization control agent is too broad in my opinion. It is especially important since the authors insist that extracellular proteins, which pAsp models in in vitro system have nothing to do with intrafibrillar mineralization. I would be also more cautious in ruling out the role of NCPs in collagen mineralization in vivo. We know that mutations in these proteins affect collagen intrafibrillar mineralization, and even in vitro different NCPs affect collagen mineralization in a different fashion (see Deshpande et al. *Biomacromolecules*, 2011, 12, 8, 2933–2945).

This is a great contribution to the field and it was a pleasure to review the manuscript.

Elia Beniash

We would like to thank the reviewers for the positive comments on our revised manuscript. The remaining points raised by the reviewers have been addressed point by point as below.

Reviewer 1

Detailed Remark 1-1

Page 36, Figure 3b: In case the scale bar is true, the triple-helices have mostly a diameter of about 1 nm showing the individual three alpha-chains, which could be resolved by X-rays. I am wondering about the discrepancy to the known value of about 1.45 nm for the triple-helix diameter from the literature. Do have the authors have an explanation for this?

Author Response

Figure R1. A zoom-in image of previous Fig. 3b (now Fig. 3b, panel ii) showing the width of collagen triple helix, which varies from ~ 1.1 nm (highlighted by the red dashed line) to ~ 1.5 nm (highlighted by the yellow dashed line). Scale bar: 1 nm.

We confirm that the scale bars in that figure are correct, and emphasize that our electron density map shows that the width of collagen triple helix is varying from ~ 1.1 nm to ~ 1.5 nm (Figure R1). This is in line with the structure of the collagen triple-helix, which is not a cylindrical rod with a uniform diameter, but more like a wormy rope with the width varying along the ~ 300 nm of its length.

Notably, that the width of collagen triple helix varies in this range was also observed in our previous X-ray analysis of the collagen triple-helix (Orgel et al., Plos One, 2014), and in the AFM (Baranauskas et al., Appl. Biochem. Biotech., 1998) and TEM (Simon et al., Sci. Rep., 2018) studies of other groups.

We have now clarified this in the text and cited the above mentioned references in Line 171-172: “The widths of the collagen triple helix molecules vary between ~ 1.1 nm to ~ 1.5 nm, in line with previous reports.^{30,33,34}”

We also noticed that the image brightness was set too low in the figure, which made the collagen triple-helix appear slightly narrower. This problem has now been fixed.

Detailed Remark 1-2

Page 8, line 175: The authors write in the description to Fig. 3 about a channel diameter of 2-3 nm, which gives the impression that it has a statistical variation over space. However, what they mean is that the individual pore diameter is (2×3) nm² as seen in Fig. 3b, thus indicating an elongated pore in cross section.

Author Response

We realize that the previous Fig. 3b is indeed misleading and apologize for the problem. Although the cross sections of the channels are rectangular at certain locations (e.g., 0.773 D, as shown in previous Fig. 3b), their shapes are generally irregular as shown in Supplementary Movie S4. However, the widths of the channels remains in the range of 2–3 nm. We have now added two more slices cut in the gap region (at 0.505 D and 0.872 D, respectively) to show the varied shapes of the channel cross sections, and have adjusted the Figure numbers, Figure caption (Line 846-853) and text in Line 177-179 accordingly. We have also modified the schemes in Figure 6 to make the channels less rectangular in order to prevent confusion.

Line 177-179 now reads:

Each of the channels is 2-3 nm in width and its cross section varies in shape along the channel (as shown by Panels ii to iv of Figure 3b, and Supplementary Movie S4).

And Line 846-853 now reads: (a) Intermolecular voids within the collagen structure, with the gap/overlap zones and the banding structure labelled. The white structures correspond to the collagen molecules, and the channels between them are labelled with different colors based on their connectivity. (b): Cross section slices viewed along the collagen fibril cut at position i (0.075 D), ii (0.505 D), iii (0.773 D) and iv (0.872 D) as highlighted by white arrows in (a), respectively. Panel i shows the typical structures of overlap region, while Panels ii to iv shows the 2-3 nm wide channels in gap region with varying cross section shapes. The unit cell is highlighted by the yellow parallelogram. A ~0.3 nm wide small connection is highlighted by magenta arrow in Panel ii.

Detailed Remark 1-3

Page 39, Figure 6a: The scheme of the principal Fig. 6a of the paper is misleading to a non-expert since it gives the impression that the macroscopic bar (collagen fibril) at top and the zoom of pores below have the same orientation with respect to each other. This would give the impression that the bar has pores oriented radial symmetric but not along the collagen fibril long axis. The authors could either indicate a cross sectional cut on the bar and a turn of 90 degrees from the bar to the zoom or they could add a bar in cross section from the beginning and indicate a zoom there.

Author Response

We have adjusted Figure 6a following the suggestion of the reviewer.

Detailed Remark 1-4

Page 6 line 125-129 and page 14, line 326-336: The authors discuss on page 6 the (low) possibility occurrence of ACP and OCP in bone. They could add for OCP case a recent paper from Simon et al. in Scientific Reports 8, 13696 (2018) additional to the Vyalikh and Mahmud papers, citations 28 and 29.

Additionally, the authors discuss the chemical templating and molecular recognition of collagen and HAP citing Silver and Xu, citations 22 and 23. Here, they could also add as citation Simon et al. Scientific Reports from 2018 since the templating is directly shown in TEM by imaging individual triple-helices mineralized by HAP on the atomic level.

Author Response

We have added this reference to the two locations suggested by the reviewer.

Reviewer #2

Detailed Remark 2-1

There seems to be a major disconnect when they argue that the crystal orientation is governed by confinement in collagen channels, but then in several locations they state that the crystals push aside the collagen molecules. And they seem to argue this occurs during lateral growth, as well as crystals pushing into the densely-packed overlap region. I think this contradictory argument could be clarified if one were to consider the confinement belonging to crystals growing within the ACP phase, and this phase is what is originally confined by the collagen during infiltration. I believe their SAXS data (which they cite as matching bone data) supports this premise.

Line 276: "This can be attributed to the formation of amorphous calcium carbonate (ACC) within the fibrils, where this causes a small decrease of the lateral packing distance of the collagen molecules. This is demonstrated by broadening of the SAXS peak at $q = 4.08 \text{ nm}^{-1}$ ($\sim 1.5 \text{ nm}$), which corresponds to a reduction in the separation of the collagen molecules (Figure 5h and 5i)."

- This broadening seems to have already taken place to a great extent before the crystallization process, where the blue curve is already broadening (as well as intensified) and the pink curve shows roughly two overlapping peaks corresponding to the initial and final state. Doesn't this suggest that the shifting around of the collagen molecules you describe takes place before the crystallization peaks emerge? (where the green and blue sharpen up as the crystals further reduce the water content and thus spacings). In other words, it seems like the collagen reorganization and thus confined space is created by infiltration of the amorphous phase. You sort of added that into your revision, except in other places you state that the crystals are pushing aside the collagen (and in your Fig. 6 schematic).

For example:

Figure 6 caption: "Nuclei with their fast growing c-axes parallel to the channels out-compete those with other orientations (d), and develop into platelets by pushing aside the collagen molecules, as shown in (e)."

Line 388: "The lateral growth of the needles will then generate platelets by pushing aside neighbouring collagen molecules (Figure 6e)."

Line 392: "Continued growth of the crystals along the c-axis will cause them to extend into the overlap region, causing a rearrangement of the collagen molecules, and a concomitant translocation of the included ACP."

Author Response

As stated in the text, although the collagen molecules are pushed apart upon ACP infiltration, they are compressed even more during the amorphous to crystalline transformation. We therefore believe that the text in our manuscript is accurate. However, following the comments of the reviewer, we have further clarified how the collagen molecules confine the HAp crystals at the early stage and induce the uniaxial orientation, but are then later pushed aside by developing crystals.

We propose the following process: at the early stage, the HAp crystals with their c-axes oriented along the channels can easily grow into needles without having to push apart the collagen molecules. The HAp crystals oriented in other directions, in contrast, have to push apart collagen molecules in order to grow longer, which is energetically less favoured (but not impossible). As a result, those misoriented HAp crystals will be consumed by the oriented crystals. After that, all of the remaining HAp crystals are oriented similarly, and they have no other choice but to push apart collagen molecules in order to grow wider/longer. We have clarified this in Line 389-391.

Line 389-391 now reads:

... while crystals that are oriented in other directions can only grow by pushing apart the collagen molecules, which is energetically less favourable...

Detailed Remark 2-2

- What does translocation of ACP mean?

Author Response

The word 'translocation' is defined in the *Oxford English Dictionary* as 'the movement of something from one place to another'. We have changed the word to 'displacement' as it is more commonly used.

Detailed Remark 2-3

Figure 2 has a typo in the second sentence. Reconstructionlateral

Author Response

Corrected.

Reviewer #3

Detailed Remark 3-1

I would only ask if possible for the authors to provide their opinion on the role of pAsp in in vitro collagen mineralization. Calling it a crystallization control agent is too broad in my opinion. It is especially important since the authors insist that extracellular proteins, which pAsp models in in vitro system have nothing to do with intrafibrillar mineralization. I would be also more cautious in ruling out the role of NCPs in collagen mineralization in vivo. We know that mutations in these proteins affect collagen intrafibrillar mineralization, and even in vitro different NCPs affect collagen mineralization in a different fashion (see Deshpande et al. *Biomacromolecules*, 2011, 12, 8, 2933–2945).

Author Response

Our previous discussions about pAsp and NCPs were indeed broad and not very rigorous. We conclude from our results that these molecules are not involved in the orientational control of intrafibrillar minerals but do not mean to imply that they have no role in intrafibrillar mineralization. As demonstrated in several *in vitro* model systems, charged polymers such as pAsp or pAH clearly play an important role in facilitating the intrafibrillar mineralization of collagen fibrils, although there are different views on the detailed mechanism. It is very well possible that NCPs play a similar role during collagen mineralization *in vivo*, as implied by the reference suggested by the reviewer. We have adjusted our text in Line 459-464 to describe our view on this issue more precisely, and the reference mentioned by the reviewer has also been added.

Line 459-464 now reads: Therefore, these non-collagenous molecules are unlikely to be involved in the orientational control of intrafibrillar crystal formation. However, they most certainly play a role in intrafibrillar mineralization by facilitating mineral infiltration,⁶⁹ as has been demonstrated for polyelectrolytes (e.g., pAsp) in *in vitro* model systems.^{15,38} These molecules may also be involved in the control of extrafibrillar mineralization, which accounts for a significant fraction of the mineral content in bone.⁷⁰

Reviewers' Comments:

Reviewer #1:

Remarks to the Author:

10th August 2020

Comments of Referee 1:

Nature Communications manuscript NCOMMS-20-11225C

"Intermolecular Channels Direct Crystal Orientation in Mineralized Collagen" by Prof. Sommerdijk and colleagues"

Resubmitted (rebuttal) from 3rd August 2020

The authors responded point by point to my questions satisfactory and added the citation required, thus I recommend to publish.

1. Page 36, Figure 3b:

The authors now indicate the variation of the triple-helix diameter from 1.0-1.5 nm.

2. Page 8, line 175:

The authors clarified the channel size.

3. Page 39, Figure 6a:

Figure changed.

4. Page 6 line 125-129 and page 14, line 326-336

The authors added the indicated paper.

Signed: Paul Simon